# Drivers of the spatiotemporal distribution of dissolved nitrous oxide and air-sea exchange in a coastal Mediterranean area

Susana Flecha[1], Mercedes de la Paz[2], Fiz Fernández Pérez[2], Núria Marbà[3], Carlos Morell[3], Eva Alou-Font[4], Joaquín Tintoré[3,5] and Iris E. Hendriks[3]

[1]Instituto de Ciencias Marinas de Andalucía (ICMAN-CSIC), Puerto Real, Cádiz, Spain.
[2]Instituto de Investigaciones Marinas (CSIC), Vigo, Spain
[3]Mediterranean Institute for Advanced Studies (CSIC-UIB), Esporles, Spain
[4]King Abdullah University of Science and Technology (KAUST), Thuwal, Kingdom of Saudi Arabia.
[5]Balearic Islands Coastal Observing and Forecasting System (SOCIB), Palma, Spain

*Correspondence to*: Susana Flecha (susana.flecha@csic.es)

**Abstract.** Among the well-known greenhouse gases (GHG), nitrous oxide ($N_2O$) is the third most impactful, possessing a global warming potential approximately 300 times greater than that of carbon dioxide ($CO_2$) over a century. The distribution of $N_2O$ in aquatic environments exhibits notable spatial and temporal variations, and emissions remain inadequately constrained and underrepresented in global $N_2O$ emission inventories, particularly from coastal zones. This study focuses on $N_2O$ levels and air-sea fluxes in the coastal waters of the Balearic Islands Archipelago in the Western Mediterranean Basin. Data were gathered between 2018 and 2023 at three coastal monitoring stations: two in the densely populated island of Mallorca and the third in the well-preserved National Park of the Cabrera Archipelago. Seawater $N_2O$ concentrations varied from 6.5 to 9.9 nmol $L^{-1}$, with no significant differences detected across the sites. When these sink/source strengths are integrated on an annual basis, the Balearic Sea is close to equilibrium with atmospheric $N_2O$, resulting in a neutral atmosphere-ocean exchange ($0.1\pm0.2$ μmol $m^{-2}$ $d^{-1}$). A consistent seasonal pattern was noted during the study period. Machine learning analysis indicated that seawater temperature was the primary factor influencing $N_2O$ concentrations, with lesser contributions from chlorophyll levels and salinity.

## 1 Introduction

Nitrous oxide ($N_2O$) is a potent greenhouse gas (GHG) with 300 times higher warming potential per mole than carbon dioxide ($CO_2$) on a 100- year time scale (Solomon et al., 2007). Atmospheric $N_2O$ levels have risen by more than 18% since preindustrial times and increased by 332 ppb between 2011 and 2019 (Masson- Delmotte et al., 2021). The ocean $N_2O$ budget is highly sensitive to climate change and significantly influences the climate system (Ravishankara et al., 2009). Variations in temperature, ocean circulation, and biological activity can alter the production and release of $N_2O$ from the oceans. Consequently, the concentration of $N_2O$ in the atmosphere influences global warming and climate patterns, creating a feedback loop between oceanic processes and the climate system. Marine $N_2O$ sources account for one-third of the natural emissions to the atmosphere, yielding a net source of 3.5 (2.5-4.7) Tg N $y^{-1}$, excluding coastal contributions (Tian et al., 2024). $N_2O$ is primarily produced through nitrification and denitrification pathways (Freing et al., 2012). Nitrification is an aerobic process in which ammonium ($NH_4^+$) is oxidized to nitrate ($NO_3^-$), producing $N_2O$ as a byproduct, favored by low-oxygen conditions. In contrast, denitrification occurs under suboxic to anoxic conditions and involves the stepwise reduction of

nitrate to dinitrogen gas ($N_2$), also releasing $N_2O$ as an intermediate. These microbial processes are
particularly sensitive to environmental gradients in oxygen, nutrients, and the availability of organic matter.
The nitrogen cycle is one of the most complex regulating factors of primary production, highly dependent
on dissolved oxygen concentration and the prevailing redox conditions (Codispoti, 2010). In coastal
environments, significant variability exists in the nitrogen cycle, where land-derived nutrient inputs, coastal
upwelling events, and complex biogeochemical processes play crucial roles (Doney, 2010), contributing
significantly to the spatiotemporal variability of the $N_2O$ concentrations (Nevison et al., 1995). In estuarine
and coastal waters, the effects of climate change may be more pronounced, such as ocean acidification (OA,
Carstensen and Duarte, 2019), **which can** enhanc**e the generation of $N_2O$** (Wan et al., 2023; Zhou et al.,
2023). Despite the necessity for a better understanding of atmospheric and oceanic inventories of non-$CO_2$
GHGs to provide realistic and accurate models for future scenarios, there are limited open ocean and coastal
monitoring time series networks compared to $CO_2$ (Bakker et al., 2014; de la Paz et al., 2015; Farías et al.,
2007; Ma et al., 2019; Wilson et al., 2017).
The Mediterranean Sea is a semi-enclosed basin surrounded by highly sensitive coastal zones, particularly
vulnerable to human activities. Factors such as high population density, widespread urbanization, and
intensive agriculture have escalated risks of pollution and habitat degradation in the region (UNEP/MAP,
2012). Due to its distinct biogeochemical and hydrodynamic features, this basin has been identified as a
"hotspot" for climate change research (Giorgi, 2006). The impacts of global warming and extreme weather
events are expected to be more severe in the Mediterranean compared to other oceanic regions (Giorgi,
2006; Giorgi and Lionello, 2008; Masson-Delmotte et al., 2021). Despite representing just 0.82% of the
global ocean surface, the Mediterranean hosts 4-18% of the world's marine biodiversity, including
numerous endemic species (Bianchi and Morri, 2000; Mouillot et al., 2011). Rising temperatures and OA
threaten the biodiversity of the region (Micheli et al., 2013). Additionally, anthropogenic pressures along
the Mediterranean coast have intensified due to rapid population growth and economic activities. In the
Western Mediterranean, high tourism and coastal development levels have left only a small fraction of the
coastline in a natural state, with even fewer areas under protection (EEA, 1999).
The Balearic Islands Archipelago, located in the Western Mediterranean, comprises the islands of Mallorca,
Menorca, Ibiza, and Formentera, with a combined coastline of 1,723 km. Renowned as a major European
tourist destination, tourism accounts for approximately 45% of the total Gross Domestic Product of the
archipelago. Visitor numbers have surged dramatically over the past century, reaching nearly 18 million in
2023 (Institut d'Estadística de les Illes Balears, Spain), compared to a resident population of around 1.2
million. Coastal ecosystems in the Balearic Islands are vital for the local economy. Meadows of the endemic
seagrass *Posidonia oceanica* extend across depths of up to 45 m in the Balearic Sea, providing critical
ecosystem services such as carbon sequestration (Duarte et al., 2005), oxygen production (Hendriks et al.,
2022), biodiversity support, coastal erosion prevention, sediment stabilization, and water transparency
(Barbier et al., 2011). However, these ecosystems face increasing pressure from recreational activities and
other anthropogenic impacts.
Given the increasing threats to these ecosystems, understanding the relationship between anthropogenic
pressures and GHG emissions, particularly $N_2O$, has become urgent. The absence of long-term $N_2O$ datasets
in the Mediterranean Sea, along with the uncertainties surrounding current emissions estimates, underscores
the importance of assessing coastal areas with varying human impacts. These evaluations are essential for
improving coastal $N_2O$ emissions estimates and refining global ocean $N_2O$ budgets. In this study, we
evaluate the spatial and temporal $N_2O$ concentrations in surface waters and the air-sea exchange in the
coastal area of the Balearic Islands Archipelago and estimate the potential drivers of the observed variability
in $N_2O$ concentrations. We focused on three different sites in the coastal zone: a highly impacted site, a
medium-impacted site, both located near the island of Mallorca, and a pristine site in the Cabrera National
Park Archipelago.

**2 Methods**

**2.1 Study area**
We collected physicochemical and biogeochemical parameters from three stations in the Balearic Sea
within the Western Mediterranean Basin (Fig. 1A), which are part of the Balearic Ocean Acidification Time
Series (BOATS; Flecha et al., 2022). The data presented here were collected from 2018 to 2023; however,
data availability varies among the different stations.
Two sampling sites are fixed monitoring stations equipped with autonomous sensors. The first station,
established in December 2018, is located in the Bay of Palma (PB: 39.492848ºN, 2.700405ºE, at a depth of
approximately 30 m, Figure 1B) and is part of the fixed monitoring network of the Balearic Islands Coastal
Observing and Forecasting System (Tintoré et al., 2019; 2013- SOCIB; https://www.socib.es/).
Temperature (ºC) and salinity (PSU) data were obtained from the SOCIB buoy sensors; see Tintoré and
Casas (2022) for details about the sensors. Additionally, a MiniDot sensor (PME, Inc©) recorded dissolved
oxygen (DO) hourly, with a manufacturer accuracy of ±5%. The sensor location and water sampling depth
were set at 1 m.
The second fixed monitoring station is located in the Bay of Santa Maria (Fig. 1B) within the Cabrera
Archipelago National Park (CA: 39.151395º N, 2.950823º E, ~8 m depth) and was installed in November
2019. It is situated in an area under governmental protection and is regarded as a pristine site with no
apparent human influence. Temperature (ºC), salinity (PSU), and DO data were collected hourly using
SBE37-SMP-ODO (Sea-Bird Scientific Electronics®) and a MiniDot. Both sensors were secured to a
mooring line at approximately 4 m depth, and water samples were taken at the same depth. The
manufacturer accuracy of measurements was ±0.002 ºC, ±0.003 mS/cm, and ±5% for temperature,
conductivity, and oxygen sensors, respectively.
At PB and CA stations, samples of dissolved $N_2O$, DO, dissolved organic carbon (DOC), Chlorophyll *a*
(Chl *a*), inorganic nutrients, including nitrate ($NO_3^-$), nitrite ($NO_2^-$), phosphate ($PO_4^{3-}$), silicate (Si $(OH)_4$),
ammonia ($NH_4$), and Total Nitrogen (TN) were collected monthly.
The third sampling point is located in the coastal area near the Cape Ses Salines lighthouse (CS: 39.2649º
N, 3.0535º E, Figure 1B). At this site, with a total bottom depth of 2 m, data were collected biweekly from
surface water directly off the coast at approximately 0.5 m depth. Sampling began in August 2018.
Temperature (ºC), salinity (PSU), $N_2O$, DOC, Chl *a*, and inorganic nutrients were collected from the same

volume of surface water. DO data were gathered using a MiniDot sensor starting in August 2018. Validation of DO sensor data was performed with DO water samples collected from PB and CA stations, as detailed in Agueda-Aramburu et al. (2024).

The three sites differ in their physical setting and benthic characteristics. PB is located in an open bay with moderate anthropogenic influence and limited seagrass coverage within the sampling area. CA, on the other hand, is located in a semi-enclosed bay within a protected marine park and is surrounded by extensive *Posidonia oceanica* meadows, which are known to influence nutrient and gas dynamics. CS is a shallow nearshore site with intermittent seagrass patches and high exposure to coastal land-sea interactions. These contrasts are relevant to the interpretation of local biogeochemical processes, including $N_2O$ production and air-sea fluxes.

## 2.2 Data collection and analysis

### 2.2.1 Biogeochemical variables

To determine $N_2O$ levels, samples were collected in duplicate using 120 mL serum vials sealed with grey-butyl rubber stoppers and aluminium crimps. After being sealed, the samples were preserved with $HgCl_2$ and stored upside-down until analysis. $N_2O$ concentrations were analyzed at the AQUANITROMET laboratory (https://www.iim.csic.es/en/about-iim/organization/aquanitromet-analysis-greenhouse-effect-gases-natural-waters) of the Instituto de Investigaciones Marinas (IIM-CSIC, Vigo, Spain), employing a static headspace equilibration technique combined with gas chromatography (GC) equipped with electron capture detection, following the methodology detailed by De la Paz et al. (2015). To create the headspace, 20 mL of nitrogen gas from a Tedlar bag at atmospheric pressure was introduced into the vials, while simultaneously extracting the same volume of water sample using a double-needle setup. The vials were shaken and left to equilibrate for at least 12 hours in a temperature-controlled environment. For injection into the GC, a brine solution was added through one needle to displace the headspace gas into the GC via the second needle.

The GC, an Agilent 7890 GC, was calibrated using three standard gas mixtures: a NOAA-certified primary standard resembling atmospheric air composition and two additional $N_2O$-in-$N_2$ mixtures supplied by Air Liquide (De la Paz et al., 2015). While participating in the first large-scale international Inter-Laboratory Comparison experiment for seawater $N_2O$ measurements (Wilson et al., 2018), an additional certified standard from the Scientific Committee for Oceanographic Research (SCOR) was utilized. The precision of the analysis was determined to be 0.5%, calculated from the average coefficient of variation across 400 replicate measurements.

To determine the DO concentration, samples were collected in five borosilicate bottles and then analyzed using the Winkler method modified by Benson and Krause (1984), employing potentiometric titration with a Metrohm 808 Titrando. The precision of the DO analysis was estimated to be less than $\pm2$ $\mu mol$ $kg^{-1}$.

Chl *a* samples were collected in glass bottles and three replicates were filtered in the laboratory using a Whatman GF/F glass fiber filter. Chl *a* extraction was performed with 90% acetone for 24 hours in dark conditions, and the samples were measured with a fluorometer (Turner Designs Instrument, Model 7200-

00). The fluorometer was calibrated with a pure Chl *a* standard derived from *Anacystis nidulans* algae, Sigma Chemical Company (Knap et al., 1996).

DOC samples were collected in two pre-combustioned borosilicate bottles and then filtered through pre-combusted Whatman GF/F glass fiber filters and stored in two pre-combusted borosilicate vials containing 25 µL of orthophosphoric acid ($H_3PO_4$). The samples were analyzed with a Shimadzu TOC-L analyzer following the method described by Álvarez-Salgado and Miller (1998), which is based on catalytic oxidation at a high temperature of 680 ºC.

Inorganic nutrients samples were collected in four 10ml Falcom vials at the same time with the DOC sample processing. Concentrations of $NO_3^-$, $NO_2^-$, $PO_4^{3-}$, Si $(OH)_4$, $NH_4$, and Total Nitrogen were obtained from the analysis laboratory at the Mediterranean Center for Marine and Environmental Research (CMIMA, Barcelona, Spain) using the Autoanalyser AA3 HR (Seal Analytical, United Kingdom) through continuous flow analysis. The precision, estimated from the coefficient of variation based on replicate analyses of the same water samples (n = 10), ranged from 0.13 to 0.5 %.

**2.2.2 Meteorological and atmospheric data**

Wind speed at a height of 10 m was provided by the Agencia Estatal de Meteorología (AEMET) from the Sant Joan Airport station in Palma, Spain (Fig. 1B). Monthly averaged data of atmospheric $N_2O$ molar fraction was obtained from the monitoring station at Lampedusa (LMP), Italy, as part of the NOAA (National Oceanic and Atmospheric Administration) monitoring network (http://www.esrl.noaa.gov/gmd/dv/site/; Lan et al. 2024).

**2.3 Flux estimation and other calculations**

To calculate the water-atmosphere $N_2O$ fluxes (µmol m$^{-2}$ d$^{-1}$) the following equation was used:

$$F = k * (Cw - C^*) \tag{1}$$

Where $k$ (cm h$^{-1}$) is the gas transfer velocity, $Cw$ is the concentration of $N_2O$ dissolved in water samples (mol$^{-1}$), $C^*$ is the gas saturation concentration calculated as the product of the atmospheric fraction of $N_2O$ and the solubility coefficient proposed by Weiss et al. (1980). To compute the fluxes of $N_2O$, we used the monthly mean atmospheric molar $N_2O$ obtained from NOAA.

The most suitable parameterization for gas transfer ($k$) in coastal areas with seagrass ecosystems, which are characterized by limited wind fetch, representing the study area in the Balearic Sea, was utilized. This equation was described by Dobashi and Ho (2023) as follows:

$$k = 0.143 \ U_{10}^2 \tag{2}$$

Emissions to the atmosphere are indicated by positive values. Additionally, due to the bottom depth differences between stations and to compare with other studies, the gas transfer parameterizations described by Wanninkhof (2014) and Cole & Caraco (1998) were also used to determine water-atmosphere $N_2O$ fluxes and differences are presented in the obtained annual $N_2O$ fluxes.

Annual $N_2O$ fluxes were calculated by integrating daily flux estimates over time for each station and year using the trapezoidal rule. For each group of samples, the area under the flux-time curve was computed based on the sampling dates, providing an estimate of the total annual air-sea flux. This method accounts

for the uneven temporal distribution of data and offers a reliable approach to derive annual fluxes from
discrete observations.
Furthermore, saturation levels for $N_2O$ (Sat%$N_2O$) are expressed as a percentage, determined by the ratio
of the observed $N_2O$ concentration to the calculated equilibrium concentration. $\Delta N_2O$ represents the
difference between the observed $N_2O$ concentrations. The Apparent Oxygen Utilization (AOU) is
calculated using the DO values along with the measured solubility of oxygen in seawater (Benson and
Krause, 1989).

**2.4 Data Analysis**
To test for differences in $N_2O$ concentrations and fluxes across regions and over temporal scales, simple
multifactorial general linear model analyses were conducted using MATLAB version 9.10.0 (R20211,
MathWorks Inc.).
Prior to model fitting, we performed a preliminary screening of all available environmental variables to
identify those with the highest potential predictive value for $N_2O$ dynamics. We tested the influence of
environmental drivers on $N_2O$ levels using the supervised Machine Learning (ML) method known as
Gradient Boosting Machine (GBM), which is based on decision tree models using a Cross-Validated
Boosting (CVB). GBM utilizes a specific implementation of an ensemble method that combines sequenced
base weak models to create a stronger one by applying gradient descent to minimize the model loss function.
This process included correlation analysis, variance inflation factors, and variable importance metrics
derived from preliminary GBM models. In each iteration, a new model adapts to the residuals of the
previously combined model. The combination of all the base models is achieved by summing their
predictions, and to prevent overestimation, a weighted learning factor is applied. The CVB technique allows
for more precise model evaluation by repeatedly using different datasheets to train and test the model. Both
GBM and CVB were implemented in Python version 3.12.3 using the XGBoost library (Chen and Guestrin,
2016). Although $NH_4^+$ and $PO_4^{3-}$ were included in the initial evaluation, they were excluded from the final
model due to either lower relative importance or missing data across sites, which would have compromised
model consistency.
The three most influential variables identified (temperature, Chl *a*, and salinity) were further analyzed to
explore their individual and combined effects using Ordinary Least Squares regression (OLS)  and
Generalized Additive Models (GAM). The correlation between the response of $N_2O$ and the resulting
dominant environmental variables (relative importance >10%) obtained from the CVB was assessed
through OLS using the Statsmodels library (Seabold and Perktold, 2010) in Python version 3.12.3, and by
employing GAM with the pyGAM library (Servén and Brummitt, 2020) in Python version 3.12.3 to address
the nonlinear dependency. This multi-step approach enabled robust variable selection and the identification
of mechanistic relationships driving $N_2O$ spatiotemporal patterns.
To determine the simple linear correlation between environmental variables and F$N_2O$, Pearson correlation
coefficients and *p*-values were calculated for each station using the Scipy library (Virtanen et al., 2020) in
Python version 3.12.3. Furthermore, a multiple linear regression analysis was performed to assess the
impact of various predictor variables on F$N_2O$ by applying an OLS.

## 3 Results and discussion

### 3.1 Environmental Variables Description

The observed seawater temperature patterns during the study period reflect the climatic characteristics of the Mediterranean region, with peak temperatures recorded in summer (June-August), notably reaching their maximum in August 2022 at 30.3°C, 29.9 °C, and 29°C in the PB, CA, and CS sites, respectively (Fig. 2A, 3C, and Table 1). Conversely, minimum temperatures recorded during winter (December-February) were 13.7°C in 2023 for PB, 13.04°C in 2023 for CA, and 13°C in 2019 for CS. Over the entire study period, average (± standard deviation) temperature values stood at 20.9±5.1, 20.7±4.9, and 20.4±5.0°C for PB, CA, and CS, respectively, with no statistical differences between sites ($p > 0.05$, Table 1), but denoted variability between months and years ($p < 0.005$, Table 1).

Surface salinity levels showed significant differences between sites ($p < 0.005$ with CS values), months ($p < 0.005$), and years ($p < 0.05$), maintaining average values of 37.5±0.2 practical salinity units (PSU) for PB, 37.4±0.2 PSU for CA, and 37.7±0.3 PSU for CS. The highest salinity levels were recorded during the summer of 2018 at PB (38.2 PSU), the winter of 2023 at CA (38.0 PSU), and during both the summer of 2018 and winter of 2023 at CS (38.2 PSU) (Fig. 2B, Table 1). The differences in the factory precision of the sensors that measure conductivity, and thus salinity, could explain the variations observed between the CS, PB, and CA stations in terms of salinity.

Wind speed measurements taken at a height of 10 meters displayed a noticeable seasonal trend (Fig. 2C). However, no statistical differences were found between years and months ($p > 0.05$). The highest average values occurred during spring (March-May), averaging around 3.7±0.4 m s$^{-1}$, and decreased during winter to 2.9±0.5 m s$^{-1}$ over the study period. Maximum peak values were recorded in December and January, reaching levels of up to 19.9 m s$^{-1}$.

DO levels showed distinct seasonal patterns across all stations, with the highest values recorded in spring (up to 300 μmol kg$^{-1}$) and the lowest during late summer, particularly at CA, reaching around 195 μmol kg$^{-1}$. Average DO concentrations were similar at PB and CA, at 233.5±22.0 and 231.8±25.3 μmol kg$^{-1}$, respectively, while limited data were available for CS. AOU values were primarily negative, indicating net photosynthetic production. Nutrient concentrations displayed spatial and temporal variability. $NO_3^-$ concentrations ranged from 0.1 to 6.3 μM, with higher values observed at CS (average 1.5±1.5 μM) compared to PB and CA. $NO_2^-$ levels were generally low (0.1–0.3 μM) but exhibited significant interannual differences. DOC showed seasonal variation, with values ranging from 52 to 116 μM, with the highest values in summer and lower concentrations in winter (Table 1).

### 3.2 Nitrous oxide and related biogeochemical variables.

Over the monitored period from 2018 to 2023, dissolved $N_2O$ concentrations exhibited a seasonal pattern ($p < 0.005$, Figure 3A), ranging from 6.5 to 9.9 nmol L$^{-1}$, with no significant differences between stations and years ($p > 0.05$). The highest concentrations were recorded during winter, averaging 9.0±0.2 nmol L$^{-1}$ (with a peak in February), followed closely by autumn (September-November) at 8.2±0.6 nmol L$^{-1}$ and spring at 7.9±0.7 nmol L$^{-1}$ (Table 1). In contrast, the lowest values were observed in summer, averaging 6.9±0.2 nmol L$^{-1}$, with the minimum recorded in August 2021 at PB. Opposite trends were noted in

Sat%N$_2$O percentages, with maximum values in summer and autumn and minimum values in winter, ranging from 93 to 116.7% (Figure 3B).

Chl *a* followed a distinct seasonal pattern (Fig. 3D, Table 1), with significant differences observed between sites, years, and months ($p < 0.005$). In January, we recorded the highest productivity with a mean Chl *a* value of $0.4\pm0.2$ µg L$^{-1}$, while May noted the lowest at $0.2\pm0.05$ µg L$^{-1}$. PB exhibited the highest levels at $0.3\pm0.2$ µg L$^{-1}$, whereas CA ($0.2\pm0.2$ µg L$^{-1}$) and CS ($0.2\pm0.1$ µg L$^{-1}$) displayed comparable concentrations. In 2019, all stations showed the strongest Chl *a* signal (Fig. 3D).

DO presents significant monthly differences observed across all stations ($p < 0.005$, Figure 3E and Table 1). During the spring, DO levels peaked at up to 300 µmol kg$^{-1}$, while at the end of the summer, minimum values around 160 µmol kg$^{-1}$ were recorded, particularly at the CA site. The average levels from 2018 to 2023 were consistent between PB and CA, at $233.5\pm22.0$ and $231.8\pm25.3$ µmol kg$^{-1}$, respectively. In contrast, only limited sensor data were available for CS (Fig. 3E, green dots). However, it is well-known that relevant oxygen production in the area is closely related to the presence of seagrass meadows, dominated by the endemic species *Posidonia oceanica,* as previously noted by Agueda-Aramburu et al. (2024) at the precise CA station location of this study.

AOU values were predominantly negative (Table 1), with average values of $-14.1\pm12.0$, $-11.9\pm15.3$, and $-15.1\pm4.9$ µmol kg$^{-1}$ for PB, CA, and CS sites, respectively. The negative AOU levels indicate the DO produced by the excess of photosynthesis compared to respiration, which aligns well with the organic matter content observed from the DOC data (Fig. 3F). In the coastal Balearic Sea, DOC showed similarities between sites and years ($p > 0.05$), with significant differences among months ($p < 0.05$). During the summer season, the highest average levels of $83.9\pm0.9$ µM were recorded, while winter exhibited the lowest at $73.8\pm1.5$ µM (Fig. 3F, Table 1). Variability in DOC data was notable in spring and autumn, with average values of $76.0\pm5.9$ and $76.6\pm3.1$ µM, respectively.

NO$_3^-$ levels ranged from 0.1 to 6.3 µM during the study period, showing no significant differences between months but significant differences between years ($p < 0.005$) and sites ($p < 0.005$). CS exhibited the highest NO$_3^-$ levels with an average of $1.5\pm1.5$ µM, followed by PB with $0.3\pm0.4$ µM, while CA had the lowest at $0.1\pm0.2$ µM (Fig. 3G, Table 1). NO$_2^-$ concentrations were significantly different between years ($p < 0.005$; Fig. 3H, Table 1) and between stations ($p < 0.05$ for CA, Table 1), with total average values of $0.03\pm0.02$, $0.05\pm0.05$, and $0.07\pm0.06$ µM for CA, PB, and CS, respectively.

The analysis of the effects of environmental and biogeochemical variables on N$_2$O concentrations through the CVB revealed only seven main parameters associated with the variability of N$_2$O concentrations (Fig. 4). These parameters, in order of importance, were temperature (65 %), Chl *a* (17 %), salinity (7%), DOC (5 %), NO$_3^-$ (4 %), and NO$_2^-$ (2 %).

Temperature values, the first variable modeling N$_2$O concentrations, exhibited a strong linear decreasing correlation with N$_2$O levels, with $p<0.005$ and a $R^2$ of 0.94 for all evaluated stations (not shown). The solubility of N$_2$O declines in warmer waters, which explains the observed maximum and minimum seasonal patterns of N$_2$O, primarily governed by the thermodynamic effects of temperature on N$_2$O concentrations. This strong correlation reflects the thermodynamic influence of temperature on N$_2$O solubility and its broader role in regulating stratification and microbial activity. The Sat%N$_2$O used here as a proxy for the biological consumption/production of N$_2$O shows a positive linear correlation with temperature ($p<0.005$

and a $R^2$ of 0.59; Fig. 5). This positive correlation is anticipated since the microbial activities of nitrification and denitrification are expected to increase with rising temperatures, leading to greater $N_2O$ production (Wu et al., 2018). However, our results indicate that the seasonality control of $N_2O$ by thermodynamic solubility surpassed the enhancement of microbial activity. The dominant role of temperature observed here aligns with previous large-scale coastal assessments (e.g., Yang et al., 2020), which identified temperature as a key variable influencing both gas solubility and microbial turnover, particularly in surface waters affected by seasonal thermal stratification.

The DO availability and the predominance of negative AOU levels, with no correlation to Sat%$N_2O$ ($p$>0.05, $R^2$=0.002), suggest that *in situ* $N_2O$ production makes a minor contribution to the observed temporal variability of $N_2O$ concentration. Overall, the low significance of predictors in the CVB analysis, such as Chl *a*, DO, DOC, $NO_3^-$, and $NO_2^-$, is consistent with limited $N_2O$ production from nitrification, which is associated with the seasonal remineralization cycle of organic matter, particularly enhanced under high-productivity conditions. This interpretation aligns with previous studies in coastal and shelf environments that also reported low $N_2O$ accumulation under oxygenated and nutrient-limited conditions (e.g., Babbin et al., 2015; Lin et al., 2016). These conditions can suppress canonical $N_2O$ production through nitrification and denitrification, emphasizing the sensitivity of microbial $N_2O$ cycling to ambient redox and substrate gradients.

DOC and Chl *a* also contribute to the availability of organic substrates and oxygen fluctuations, shaping microbial $N_2O$ dynamics, especially under stratified or high-productivity conditions. In fact, DOC enrichment has been linked to elevated heterotrophic activity and subsequent shifts in nitrogen cycling pathways in productive coastal areas (Froehlich et al., 2021), potentially influencing the balance between $N_2O$ production and consumption. Microbial production by nitrification and denitrification are generally considered the dominant pathways of $N_2O$ biological production in the shelf sea and open ocean (Burgos et al., 2017; Chen et al., 2021; de la Paz et al., 2024; Sierra et al., 2020). However, in the oxygenated coastal Balearic Sea waters, the Balearic Sea, near-equilibrium values for Sat%$N_2O$ and low $NO_3^-$ and $NO_2^-$ concentrations (Figs. 3B, G-H, 5, Table 1) suggest a negligible contribution from pelagic nitrification. Nevertheless, $NO_3^-$ variability may still indicate diffuse anthropogenic inputs, potentially enhancing local nitrification under favorable conditions. While benthic $N_2O$ fluxes were not measured during this study due to logistical constraints, their contribution cannot be dismissed. A follow-up study is being conducted to evaluate potential sediment-derived $N_2O$ inputs at the same study sites.

In addition, the CBV analysis identifies Chl *a* as the second driver explaining the observed variability in $N_2O$ concentrations the seasonal cycle of Chl *a* is closely coupled to that of $N_2O$, with maximum values in winter and minimum values in summer. A hypothetic mechanism that could be contributing to the $N_2O$ concentration variability in our study site is the poorly described production of $N_2O$ by epipelagic photosynthetic organisms in the light through NO reduction. Given the low $NO_3^-$ and $NO_2^-$ concentrations, the lack of correlation between AOU and $N_2O$, and the seasonal coupling between Chl *a* and $N_2O$ peaks, we propose that an additional, non-canonical pathway may contribute to $N_2O$ production in our system. Recent studies have identified a light-dependent mechanism in which marine photosynthetic organisms, including green algae and cyanobacteria, reduce intracellular nitric oxide (NO) to $N_2O$, potentially as a detoxification strategy under fluctuating nitrogen conditions or during photosynthetic stress. This NO

reduction pathway involves the activity of flavodiiron proteins (FLVs) or other reductases and may become
relevant in oxygenated, nutrient-depleted surface waters with active phytoplankton populations (Burlacot
et al., 2020). While our dataset does not allow for direct confirmation of this process, the observed seasonal
patterns and weak traditional $N_2O$ signals warrant further consideration of this alternative mechanism,
especially in coastal systems dominated by epipelagic productivity.
Finally, significant unexpected negative linear correlations ($p<0.05$) were found between $N_2O$
concentrations and silicate only at the PB and CA sites (Table 1), with correlation coefficients of -0.41 and
-0.34, respectively. This relationship and the differences in salinity values between PB and CA compared
to CS may indicate the presence or absence of groundwater input pulses in these areas (Basterretxea et al.,
2010; Sospedra et al., 2018), which can affect $N_2O$ concentrations in specific coastal regions (Calvo-Martin
et al., 2024). Since this relationship was only observed at two of the three stations considered, this parameter
was excluded from the CVB analysis. Together, these factors illustrate a multifactorial control of $N_2O$
production and accumulation in the coastal Balearic Sea, where solubility, productivity, and nutrient inputs
act in combination.
An OLS regression was conducted to evaluate the relationship among the three variables with the highest
relative importance for $N_2O$ concentrations (Importance >5%), accounting for 89% of the variability
(Temperature, Chl $a$, and salinity, Fig. 4) identified in the CVB analysis. The resulting equation is: $N_2O =$
7.42 -0.18 * Temperature + 0.28 * Chl $a$ + 0.11 * Salinity ($R^2 = 0.94$), with a $p$-value<0.005 for temperature
and $p$-values of 0.07 for Chl $a$ and 0.152 for salinity.
To resolve the nonlinear dependency of Chl $a$ and salinity on $N_2O$ concentrations, a GAM was applied,
explaining 96 % of the response variable with a $p$-value <0.005 for temperature, <0.05 for Chl $a$, and 0.1390
for salinity (Supplementary figures). The non-significance of salinity may be due to the fact that GAM
smooth terms are designed to capture broader, nonlinear effects. In this case, salinity may not have a robust
and smooth effect, or its impact may be masked by other variables, such as temperature. Salinity also
influences stratification dynamics and can indirectly affect oxygen availability and microbial processes
linked to $N_2O$. The GAM residuals ranged from -0.6 to 0.6 nmol L$^{-1}$ (Supplementary figures).
When compared to other temperate and subtropical coastal regions, our results support the emerging view
that $N_2O$ dynamics in seagrass-dominated systems may diverge substantially from classic estuarine and
shelf models, with solubility and light-driven processes playing an increasingly relevant role (Rosentreter
et al., 2023; Al-Haj et al., 2021). Placing the obtained data in a global ocean context, specifically among
studied coastal areas, the Balearic Sea presents $N_2O$ concentrations similar to those found in 2008 in the
Bohai Sea, ranging from 7.14 to 8.32 nmol L$^{-1}$ (Gu et al., 2022), but are low in comparison to upwelling
and estuarine coastal areas. The maximum $N_2O$ concentrations found in this study represent approximately
baseline levels for the other coastal and open ocean areas evaluated previously (Bange, 2006; Bange et al.,
1996; de la Paz et al., 2024; Yang et al., 2009; Zhang et al., 2008).

**3.3 Air-sea $N_2O$ exchange**
$FN_2O$ values ranged from -0.3 to 0.6 µmol m$^{-2}$ d$^{-1}$ during the study period, indicating that the coastal
Balearic Sea functions as a minor sink or is nearly in equilibrium in winter and spring and acts as a light
source in summer and autumn (Fig. 6; $p < 0.005$), closely following the seasonal temperature variations
(Fig. 6, grey line). No differences between years were observed. However, significant differences were
noted between sites, with PB exhibiting the highest $FN_2O$ levels and CA showing the lowest ($p < 0.05$).
In the PB site, positive linear correlations were found between $FN_2O$ and temperature ($R^2 = 0.67$; $p < 0.005$)
and between $FN_2O$ and $\Delta N_2O$ ($R^2 = 0.96$; $p < 0.005$). In the OLS analysis, wind speed also appears to exert
a positive influence on $FN_2O$ values, detectable only in PB data (OLS, $R^2 = 0.94$, $p < 0.05$). This effect is
likely attributable to PB site geographical characteristics, as it represents an open bay, in contrast to the
enclosed CA station in the Santa Maria Bay sampling site and the shallow coastal waters of CS.
$FN_2O$ in the most pristine station CA showed positive correlations with temperature ($R^2 = 0.79$; $p < 0.005$),
$\Delta N_2O$ ($R^2 = 0.97$; $p < 0.005$) and the $Sat\%N_2O$ ($R^2 = 0.97$; $p < 0.005$).
The CS shallow site showed positive linear correlations with temperature ($R^2 = 0.73$; $p < 0.005$), $\Delta N_2O$ ($R^2$
$= 0.98$; $p < 0.005$), and $Sat\%N_2O$ ($R^2 = 0.97$; $p < 0.005$). Salinity was also a predictive variable included in
the OLS analysis (OLS, $R^2 = 0.96$, $p < 0.05$) and was negatively related. This characteristic may be
associated with the differences in CS salinity values compared to the other stations.
Annual average fluxes for the three sites were calculated; when monthly data were unavailable, linear
interpolation was applied (Table 2). The different parameterizations for the gas transfer velocity indicate
higher values using Cole & Caraco (1998), ranging from 8.7 to 204.2 $\mu mol\ m^{-2}\ y^{-1}$, followed by results
obtained with the Wanninkhof (2014) equation, which range from 9.9 to 190.1 $\mu mol\ m^{-2}\ y^{-1,}$ and the lowest
values defined by Dobashi & Ho (2023), ranging from 5.4 to 107.4 $\mu mol\ m^{-2}\ y^{-1}$ (Table 2). These differences
are clearly influenced by the environment since the Dobashi & Ho (2023) equation was derived from studies
conducted in shallow areas at approximately 3.5 m depth, while Wanninkhof (2014) and Cole & Caraco
(1998) focused on deeper ocean areas and a lake, respectively. However, since the benthic zone of this
study is heavily covered with *P. oceanica* seagrass meadows and the fetch area is limited, we opted to retain
Dobashi & Ho (2023) for the primary results.
Similar low air-sea $N_2O$ fluxes have also been reported in shallow vegetated environments, such as
seagrass-dominated coastal lagoons in Australia (Rosentreter et al., 2023), where negative or near-zero
fluxes reflect the combined effects of high primary production, oxygenation, and limited nitrogen
availability. In contrast, higher fluxes, often exceeding 1 $\mu mol\ m^{-2}\ d^{-1}$, are typical of eutrophic estuaries or
upwelling regions (Bange, 2006; Zhang et al., 2008), highlighting the relatively pristine nature of the
Balearic coastal zone.
The results showed that all the stations are significantly neutral regarding $N_2O$ emissions to the atmosphere.
However, notably higher values were detected at the PB station, as seen in the daily $FN_2O$ values (Fig. 6).
PB exhibited significant interannual variability during the study period (Table 2). The CS and CA sites
followed the same trend as the PB station but had considerably lower values, particularly for CA. Observed
annual $N_2O$ fluxes in the Balearic Sea are much lower than the expected ranges for this ocean region, as
described by Resplandy et al. (2024) based on global reconstruction products from 1985 to 2018 (Yang et
al., 2020). These differences may relate to the substantial variability in local $N_2O$ fluxes in coastal areas
with vegetation. Coastal regions with seagrass meadows are expected to exhibit the lowest $N_2O$ flux ranges
(Rosentreter et al., 2023). These areas are considered negligible sources (Al-Haj et al., 2021) or sinks in
near-pristine seagrasses, leading to significantly biased coastal global estimations (Chen et al., 2022). These
low fluxes can be partly attributed to the biogeochemical effects of seagrass meadows. Furthermore,
elevated rates of primary production reduce the availability of inorganic nitrogen in the water column,
thereby limiting $N_2O$ generation. Additionally, the dense root systems and organic exudates of seagrass
meadows support microbial communities that further facilitate $N_2O$ consumption under oxic conditions
(Rosentreter et al., 2023; Al-Haj et al., 2021). Collectively, these mechanisms contribute to the role of
vegetated coastal systems as potential $N_2O$ sinks.
These observations place the Balearic Sea at the lower end of the global $N_2O$ flux spectrum for coastal
systems. For example, in the East China Sea and Pearl River Estuary coastal areas, fluxes ranged from 5 to
17 $\mu mol\ m^{-2}\ d^{-1}$ (Zhang et al., 2008; Lin et al., 2016), often driven by hypoxia and strong anthropogenic
nitrogen inputs—conditions absent in our study area.
Considering that European seagrasses represent approximately 6% of global seagrass cover, and that most
of this is found in the Mediterranean Sea (Jordà et al., 2012), it is crucial to understand more about the
drivers of $N_2O$ formation, consumption seasonality, and transport pathways. Additionally, increasing the
number of observations is vital since $N_2O$ data in the Mediterranean Sea is limited. Furthermore, we must
consider the ongoing coastal eutrophication linked to anthropogenic inputs from estuaries, sewage
discharge from densely populated coastal areas, and industrial effluents that may significantly alter seagrass
habitats, along with current emissions of $N_2O$ from coastal zones (Bakker et al., 2014).
The low $N_2O$ fluxes observed in our study are in line with recent assessments that emphasize the
underestimated climate mitigation potential of seagrass meadows and vegetated coastal habitats (Al-Haj et
al., 2021; Chen et al., 2022). Therefore, understanding the ecological conditions that suppress $N_2O$
emissions in these systems is crucial for refining regional and global GHG budgets.
As coastal $N_2O$ emissions are assumed to offset 30-58% of the net $CO_2$ coastal uptake radiative effect
(Resplandy et al., 2024), estimating the water $N_2O$ emission-based Global Warming Potential (GWP) can
provide a clear picture by comparing it with existing data on $CO_2$ and methane fluxes in the area. By
following the Intergovernmental Panel on Climate Change (IPCC) Assessment Report 6 (Arias et al., 2021)
updated 100-year GWP, we obtained the $GWP_{N2O}$ (i.e., the 100-year time-integrated radiative forcing from
the instant release of 1 kg of $N_2O$ is 273 times larger than the forcing of 1 kg of $CO_2$).
In the coastal Balearic Sea, considering an area 1 km offshore and 1,428 km of coastal longitude, the
average $GWP_{N2O}$ obtained in this study is $8.1 \times 10^{-7} \pm 5.7 \times 10^{-7}$ Pg $CO_2$-eq $y^{-1}$ for all sites from 2020 to
2022, which was the period with the most available data. This area estimate was used as a first-order
approximation of the active $N_2O$ exchange zone in the Balearic region, following a similar approach to
other regional-scale studies (e.g., Resplandy et al., 2024; Rosentreter et al., 2023). It encompasses both
vegetated and non-vegetated shallow coastal habitats and is not intended to represent specific benthic types.
Therefore, it reflects a conservative estimate of the total coastal footprint, acknowledging that further
habitat-specific refinement would improve spatial resolution. The associated uncertainty in the GWP
estimate incorporates the observed variability in fluxes across sites and seasons.

### 473     4 Conclusions

$N_2O$ concentrations in seawater during the study period ranged from 6.5 to 9.9 nmol $L^{-1}$ without significant
differences between the three sampling sites. Several drivers dominated the variability of $N_2O$
concentration, with temperature as the most essential factor and less critical Chlorophyll *a* and salinity.
Even with the possible biological implications in $N_2O$ formation, atmospheric forcing may control the
surface concentrations in this area. Averaged estimated $N_2O$ fluxes oscillated between -0.3 and 0.6 µmol
$m^{-2}$ $d^{-1}$, with the most influenced stations showing the highest $N_2O$ fluxes. Some sites appear to be weak
sources of $N_2O$, although many fluxes are close to equilibrium and not significantly different from zero,
following a robust seasonal pattern throughout the sampling period. Reducing anthropogenic pressures in
coastal regions is crucial to preserving marine ecosystems, as increasing human impacts, such as pollution,
overfishing, and habitat destruction, can lead to irreversible damage, loss of biodiversity, and the
degradation of vital ecosystem services. If these pressures continue to escalate, future projections indicate
more frequent and severe environmental crises, including increased coastal erosion, declining fish stocks,
and heightened vulnerability to climate change, threatening marine life and human coastal communities.
There is a strong need to enhance observations of the GHG $N_2O$ in coastal areas to better understand
dominant drivers and make more accurate predictions of future consequences under additional
anthropogenic impacts. Moreover, the values presented for the coastal Balearic Sea will aid in improving
global $N_2O$ emissions budgets in coastal vegetated areas overall.

**Data availability**
Data was obtained from the Metocean Data Repository of the Balearic Islands Coastal Observing and
Forecasting System (SOCIB). In 2024, data from the instruments on the Palma Bay Station platform
https://apps.socib.es/data-catalog was consulted on 01-30-2024. Data is also available through Tintoré &
Casas (2022) and Hendriks et al. (2023, 2025).

**Supplement link**

**Author contribution**
IE, JT, and SF conceptualized the research, data acquisition approach, and methodology. IE, SF, MdP, CM,
and AEF collected the samples and conducted the measurements. SF, MdP, and IE analyzed the data, while
SF wrote the manuscript draft. All authors reviewed and edited the manuscript.

**Competing interests**
The authors declare no competing financial interest.

**Acknowledgments**
We express our gratitude to the Cabrera National Park staff for facilitating the work done during this study,
and to the Balearic Islands Coastal Observing and Forecasting System (SOCIB) for their invaluable
assistance and the use of their fixed station in the Bay of Palma. We extend our thanks to Lidia Cucala
(IMEDEA), Andrea Carbonero, and Juan Martínez Ayala (SOCIB) for their support with sample collection
and analysis. This work contributes to CSIC Thematic Interdisciplinary Platform PTI OCEANS+. We
appreciate the AQUANITROMET service of Instituto Investigaciones Marinas for conducting the methane
analyses, and we thank the Agencia Estatal de Meteorología (AEMET) for providing the meteorological
data.
**Financial support**
Funding for this work was provided by the Spanish Ministry of Science (SumaEco, RTI2018–095441-B-
C21, CYCLE, PID2021-123723OB-C21), the Government of the Balearic Islands through la Consellería
d'Innovació, Recerca i Turisme (Projecte de recerca científica i tecnològica SEPPO, PRD2018/18), and the
2018 call of the BBVA Foundation "Ayudas a equipos de investigación científica" for the Posi-COIN
Project. SF acknowledges the financial support of the "Margalida Comas-2017" and "Vicenç Munt
Estabilitat-2022" postdoctoral contracts, as well as project AAEE111/2017 from the Balearic Islands
Government. SF is staff hired under the Generation D initiative, promoted by Red.es, an organisation
attached to the Ministry for Digital Transformation and the Civil Service, for the attraction and retention of
talent through grants and training contracts financed by the Recovery, Transformation, and Resilience Plan
through the European Union's Next Generation funds. MP acknowledges the financial support during the
study period to the contracts financed by the Spanish Ministry of Science CTM2015–74510-JIN and
PTA2019–017983-I. F.F Pérez was supported by the FICARAM+ project (PID2023‑148924OB‑l00).

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

**Figure 1**: (A) Map of the stations location in the Western Mediterranean Sea Basin and (B) detailed location
of the Bay of Palma (PB; blue dot), the Cabrera National Park (CA, red dot), and Cape Ses Salines (CS;
green dot) study sites and Palma Airport wind station (black star). Dashed lines represent bathymetry levels.
Maps were developed with the Python software version 3.12.3.

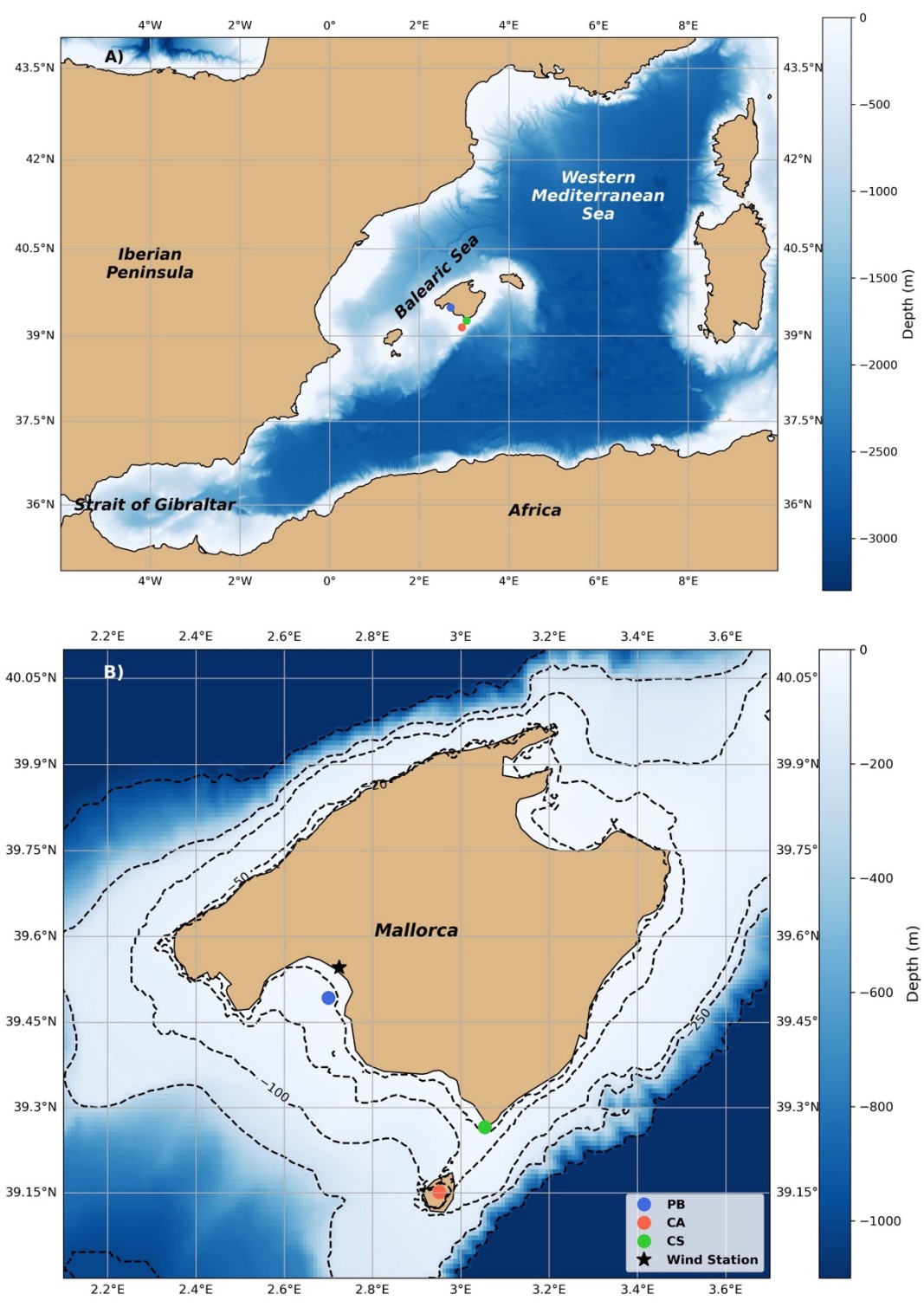



**Figure 2**: Time-series of A) daily averaged temperature (ºC) for PB (blue dots) and CA (red dots) stations,
as well as instantaneous values for the CS site (green dots); B) daily averaged salinity (PSU) for the PB
(blue dots) and CA (red triangles) stations, along with instantaneous values for the CS site (green squares);
and C) daily averaged wind speed (m s⁻¹) at 10 m height from the Palma station (Spain). Data for PB, CA,
and CS were collected at depths of 1, 4, and 0.5 m, respectively. Figures were created using Python software
version 3.12.3.



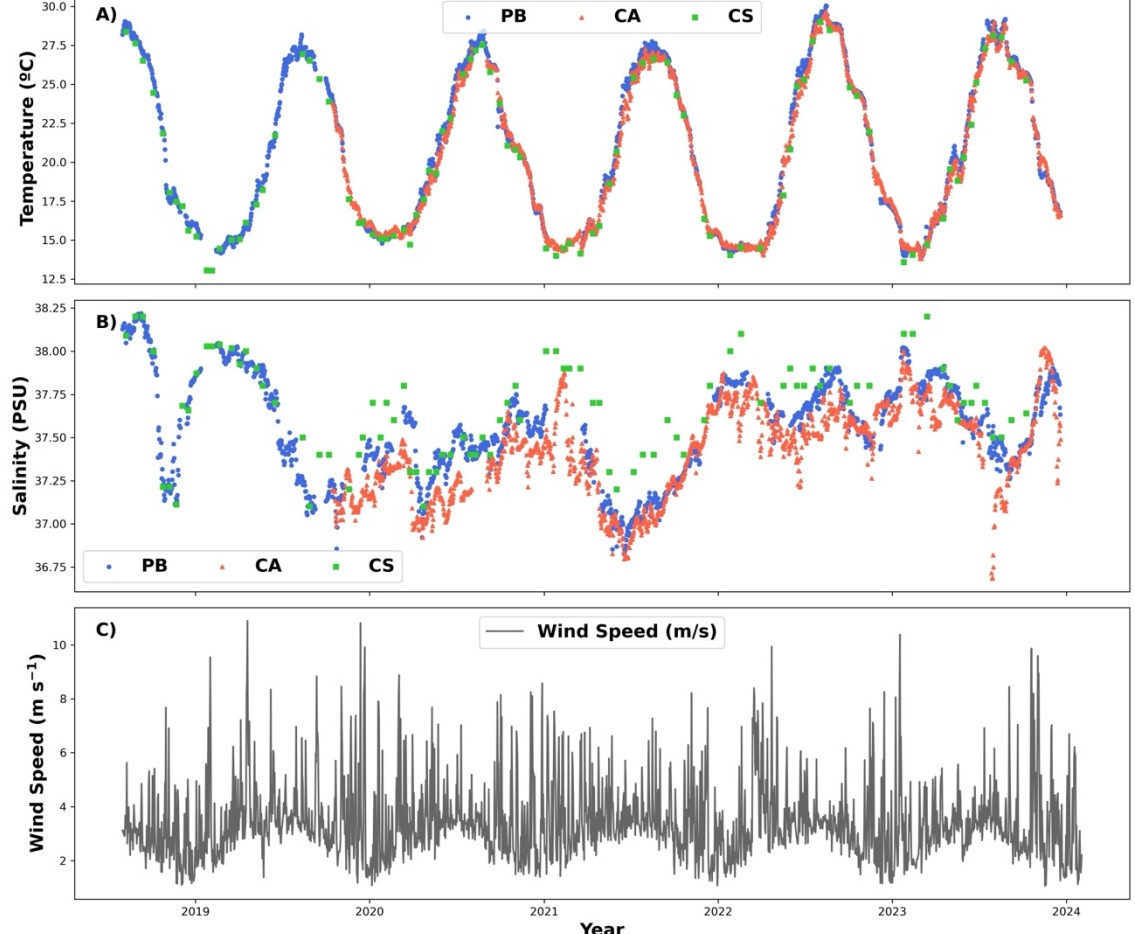






**Figure 3**: Time-series of A) Nitrous Oxide ($N_2O$) concentrations in nmol $L^{-1}$, B) $N_2O$ saturation percentage
(Sat%$N_2O$, %), C) Temperature (ºC), D) Chlorophyll *a* (Chl *a*) in µg $L^{-1}$, E) Dissolved Oxygen (DO) in
µmol $Kg^{-1}$,  F) Dissolved Organic Carbon (DOC) in µM, G) Nitrate ($NO_3^-$) in µM and H) Nitrite ($NO_2^-$) in
µM measured in the samples collected in PB (blue dots), CA (red triangles) and CS (green squares). Data
for PB, CA, and CS were collected at depths of 1, 4, and 0.5 m, respectively. Figures were developed with
the Python software version 3.12.3.

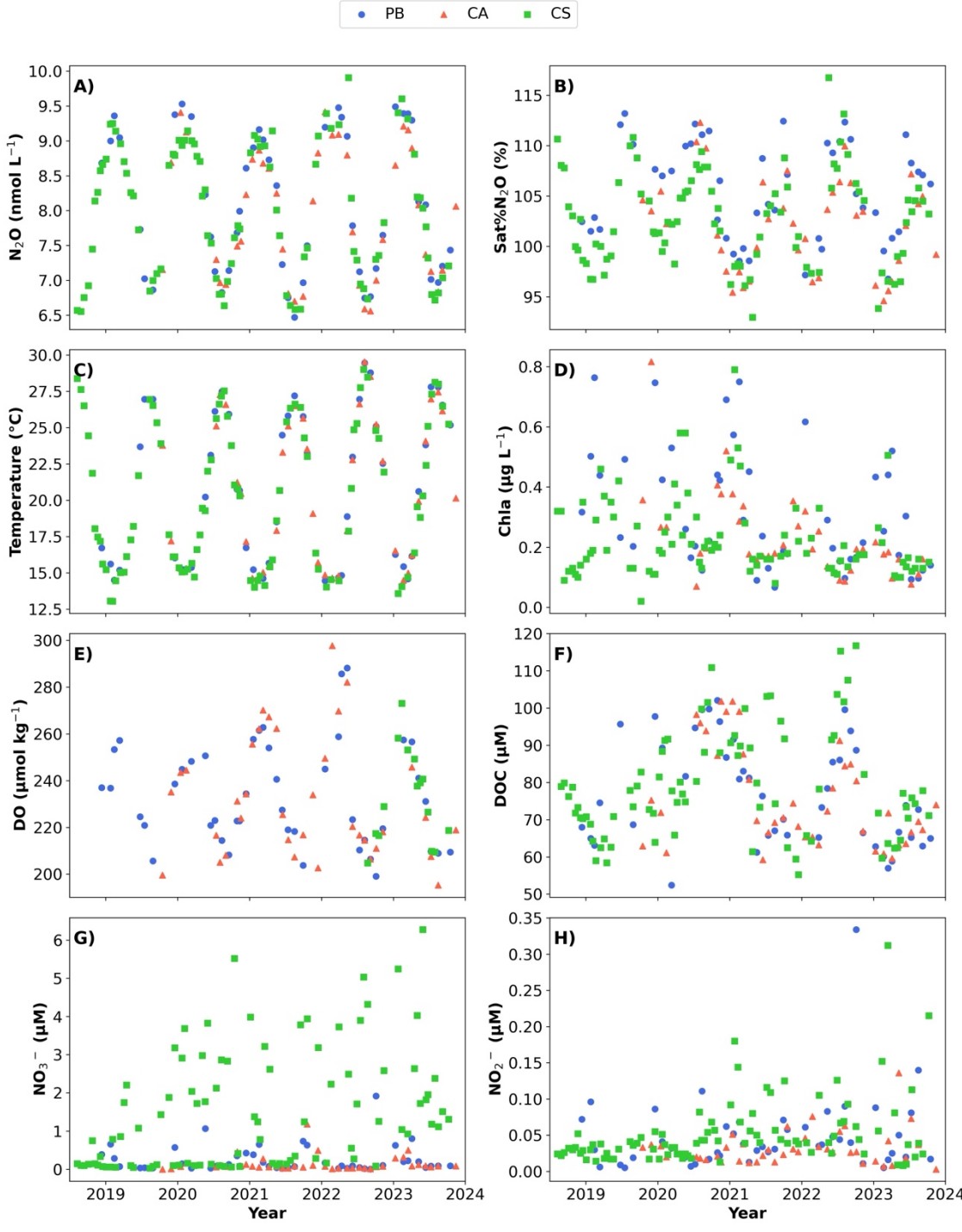


**Figure 4**: The relative importance of the leading environmental and biogeochemical variables driving the
variability of nitrous oxide concentrations variability analyzed through cross-valued boosting (CVB). These
are, in decreasing order, Temperature, Chlorophyll *a* (Chla), Salinity, Dissolved Organic Carbon (DOC),
Nitrate ($NO_3$), and Nitrite ($NO_2$). The figure was developed using Python software version 3.12.3.

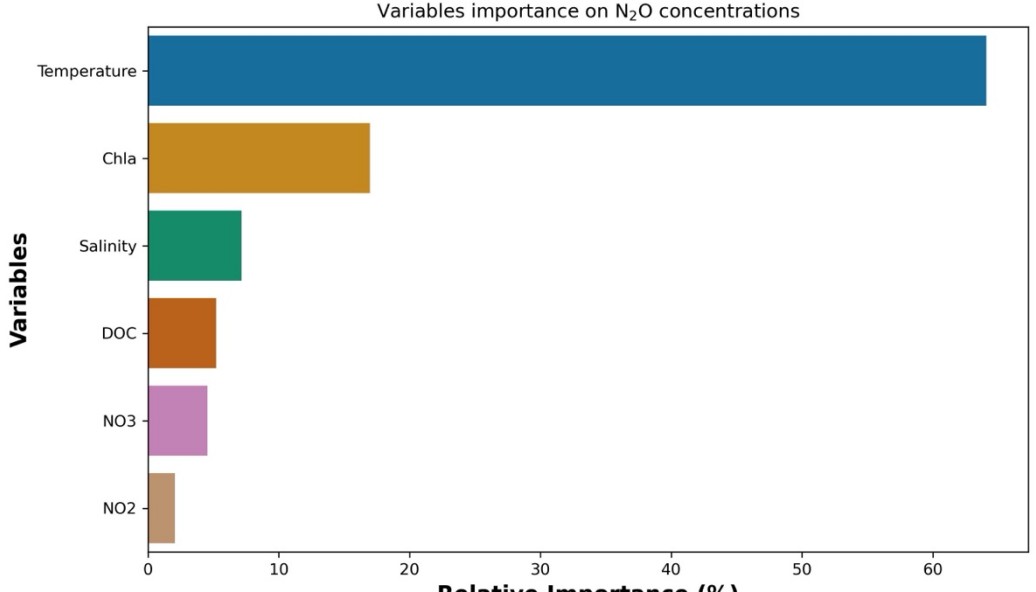

















**Figure 5**: Linear relation Saturation Percentage of $N_2O$ (Sat%$N_2O$) ) versus Temperature (ºC)  from the
data obtained in PB (blue dots), CA (red triangles) and CS (green squares). Linear equations are represented
in blue for PB, red for CA and green for CS stations. Figures were developed with the Python software
version 3.12.3.

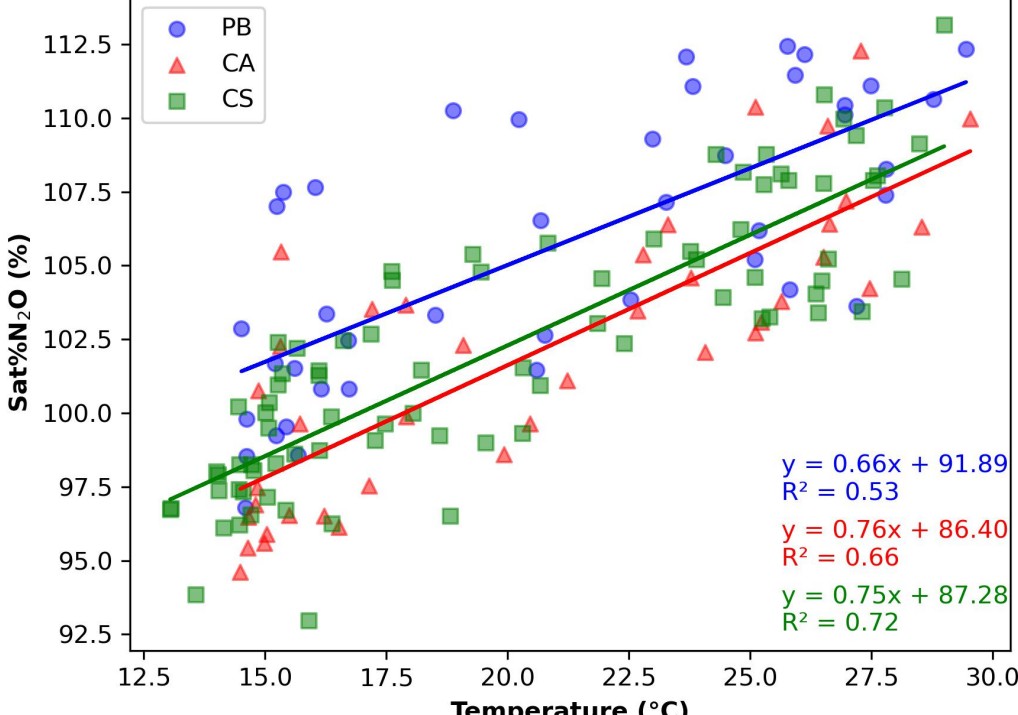
















**Figure 6**: Bar chart of monthly averaged values of air-sea Nitrous Oxide ($N_2O$) transfer ($FN_2O$) in µmol $m^{-2}$ $d^{-1}$ for all study years (left axis) and averaged temperature (ºC) values (right axis; grey line) for the PB (upper plot), CA (middle plot), and CS (lower plot) sites. Error bars represent ± standard deviation. Analysis of individual $FN_2O$ flux measurements revealed that none significantly differed from zero ($p > 0.5$). Figures were developed using Python software version 3.12.3.

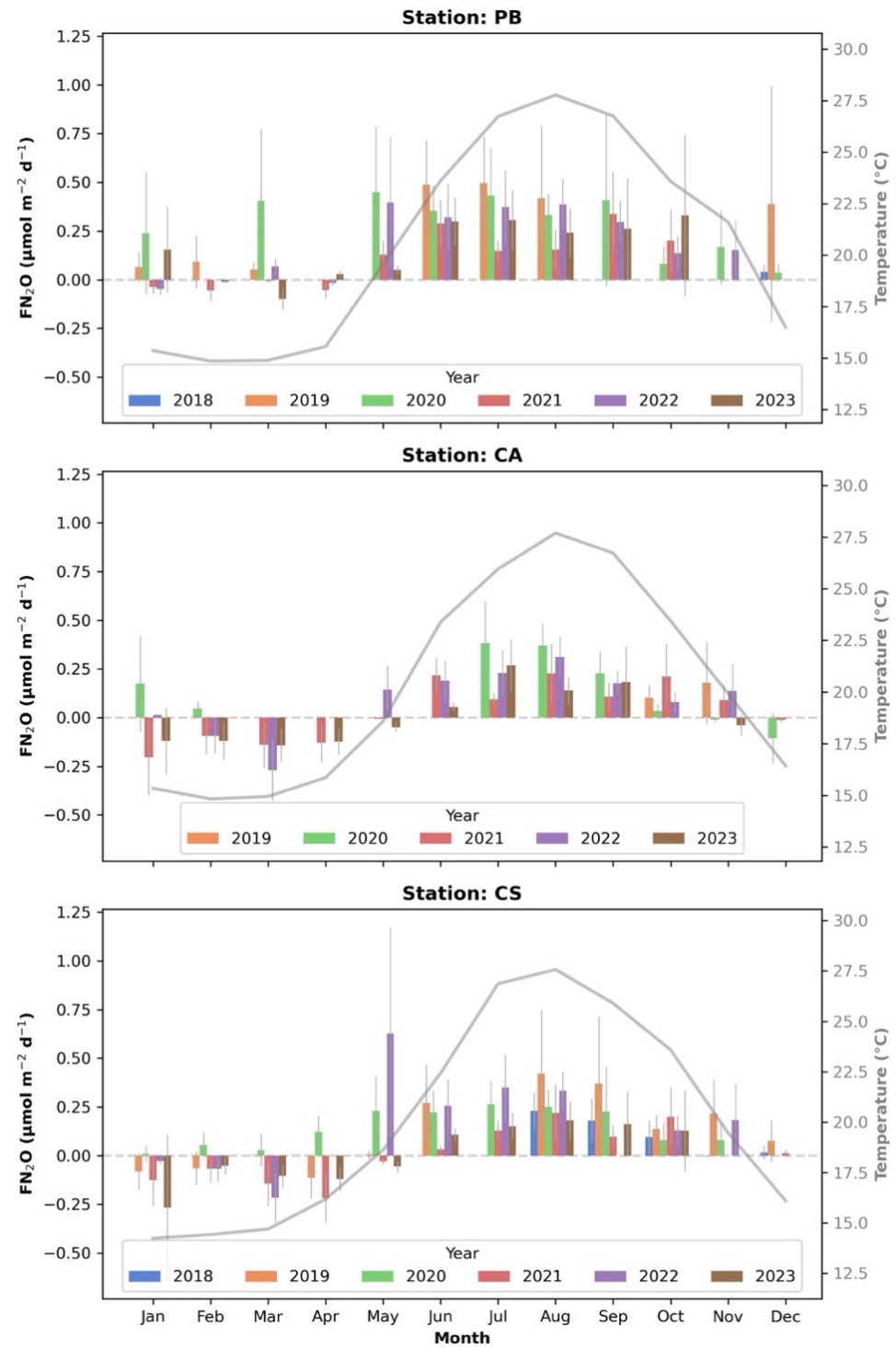

867

868

*Table 1. Seasonally averaged values of temperature (ºC), salinity (PSU), chlorophyll a (Chl a; µg L$^{-1}$),*

*dissolved oxygen (DO; µmol kg$^{-1}$), apparent oxygen utilization (AOU; µmol kg-1), nitrate (µM), nitrite*

*(µM), silicate (µM), dissolved organic carbon (DOC; µM), N$_2$O (nanomol L$^{-1}$), and % saturation of N$_2$O*

*measured in samples collected in PB (Palma Bay), CA (Cabrera National Park), and CS (Cape Ses*

*Salines). Variability is represented as ± standard deviation. * DO data is only available from August 2022*

*at the CS station. N$_2$O saturation percentages significantly differ from 100%, as evaluated by a t-test*

*(p>0.005).*

| | PB (n = 48) | | | | CA (n = 42) | | | | CS (n =91) | | | |
|---|---|---|---|---|---|---|---|---|---|---|---|---|
| | Winter | Spring | Summer | Fall | Winter | Spring | Summer | Fall | Winter | Spring | Summer | Fall |
| Temperature (°C) | 15.5±0.8 | 16.6±2.3 | 26.0±2.0 | 24.5±2.6 | 15.3±0.9 | 16.5±1.9 | 25.9±2.0 | 23.1±3.3 | 14.9±1.0 | 16.9±1.9 | 25.7±2.5 | 23.0±2.9 |
| Salinity (PSU) | 37.6±0.2 | 37.7±0.3 | 37.4±0.2 | 37.5±0.2 | 37.6±0.2 | 37.4±0.2 | 37.3±0.3 | 37.4±0.2 | 37.8±0.2 | 37.7±0.3 | 37.6±0.3 | 37.6±0.3 |
| Chl $a$ (µg L⁻¹) | 0.6±0.2 | 0.3±0.1 | 0.2±0.1 | 0.2±0.1 | 0.3±0.1 | 0.2±0.1 | 0.1±0.0 | 0.3±0.2 | 0.3±0.2 | 0.3±0.2 | 0.2±0.1 | 0.2±0.1 |
| DO (µmol Kg⁻¹) | 246.7±10.1 | 258.5±15.7 | 218.1±7.4 | 211.5±9.0 | 248.7±26.8 | 262.4±15.0 | 213.4±9.0 | 218.4±11.8 | 265.6±10.4* | 244.0±6.8* | 212.6±9.6* | 220.4±5.8* |
| AOU (µmol Kg⁻¹) | -5.5±7.9 | -23.0±16.1 | -17.8±5.3 | -5.4±5.5 | -6.6±24.8 | -26.5±13.5 | -12.4±6.9 | -6.5±7.4 | -17.3±12.1* | -13.9±4.0* | -16.7±4.9* | -13.9±2.3* |
| Nitrate (µM) | 0.2±0.4 | 0.2±0.3 | 0.1±0.0 | 0.4±0.6 | 0.2±0.2 | 0.1±0.2 | 0.1±0.0 | 0.2±0.3 | 1.4±1.6 | 1.9±1.6 | 1.4±1.5 | 1.4±1.5 |
| Nitrite (µM) | 0.06±0.03 | 0.03±0.02 | 0.05±0.04 | 0.07±0.1 | 0.03±0.02 | 0.04±0.04 | 0.03±0.02 | 0.03±0.02 | 0.05±0.05 | 0.05±0.06 | 0.05±0.03 | 0.05±0.04 |
| Silicate (µM) | 2.9±2.8 | 4.44±3.6 | 6.7±5.5 | 8.1±5.3 | 4.0±3.1 | 3.1±2.3 | 7.6±6.8 | 5.1±3.9 | 0.9±0.9 | 2.0±2.6 | 1.4±1.7 | 0.8±0.9 |
| DOC (µM) | 76.5±14.2 | 65.9±10.7 | 80.8±13.2 | 81.1±16.3 | 75.4±17.3 | 71.0±9.5 | 76.6±13.8 | 78.3±11.7 | 74.0±12.6 | 71.8±10.5 | 88.0±14.3 | 84.1±14.4 |
| N₂O (nmol L⁻¹) | 9.2±0.3 | 9.0±0.5 | 7.20±0.5 | 7.4±0.4 | 9.0±0.4 | 8.7±0.4 | 7.1±0.3 | 7.4±0.6 | 9.0±0.3 | 8.7±0.5 | 7.1±0.5 | 7.4±0.5 |
| N₂O saturation (%) | 101.8±3.3 | 102.6±4.4 | 109.6±2.8 | 107.3±3.2 | 98.6±3.4 | 97.9±2.7 | 106.6±3.2 | 103.8±3.0 | 99.0±2.0 | 100.1±4.9 | 106.9±2.9 | 104.6±2.5 |



*Table 2. Annual N₂O flux (μmol m⁻² y⁻¹) per station since 2019 for PB (Buoy of Palma) and CS (Cape Ses*
*Salines) stations, and from 2020 for the CA (Cabrera National Park) site, obtained using the Dobashi &*
*Ho, 2023; Wanninkhof, 2014; and Cole & Caraco, 1998 gas transfer parameterizations. Variability is*
*represented as ± propagated error.*

| | | Annual N$_2$O flux (μmol m$^{-2}$ y$^{-1}$) | | |
|---|---|---|---|---|
| **Station** | **Year** | **Dobashi & Ho, 2023** | **Wanninkhof, 2014** | **Cole & Caraco, 1998** |
| **PB** | 2019 | 105.4±42.1 | 186.7±74.4 | 195.7±32.1 |
| | 2020 | 107.4±25.5 | 190.1±45.2 | 204.2±20.2 |
| | 2021 | 32.4±7.0 | 57.4±12.4 | 68.2±5.9 |
| | 2022 | 60.2±10.0 | 106.8±17.7 | 124.0±8.2 |
| | 2023 | 44.6±12.3 | 79.1±21.7 | 91.3±9.6 |
| **CA** | 2020 | 58.4±17.7 | 103.4±31.3 | 128.6±14.3 |
| | 2021 | 15.0±7.7 | 26.6±13.7 | 31.7±6.3 |
| | 2022 | 23.6±7.5 | 42.0±13.2 | 58.0±6.0 |
| | 2023 | 7.3±8.8 | 13.1±15.6 | 12.0±7.0 |
| **CS** | 2019 | 45.8±13.1 | 81.2±23.3 | 86.5±10.4 |
| | 2020 | 48.4±6.6 | 85.8±11.7 | 98.2±5.3 |
| | 2021 | 6.3±6.2 | 11.1±11.0 | 16.5±5.2 |
| | 2022 | 49.2±15.1 | 87.3±26.7 | 113.3±12.1 |
| | 2023 | 5.4±6.3 | 9.7±11.1 | 8.7±4.9 |
