# Peer review of "Drivers of the spatiotemporal distribution of dissolved"

_EGUsphere, 2024_

## Referee Comment (RC2)

Reviewer's comments for Drivers of the spatiotemporal distribution of dissolved nitrous oxide and air-sea exchange in a coastal Mediterranean area

This study investigates the spatiotemporal distribution of dissolved $N_2O$ and air-sea exchange in the coastal waters of the Balearic Islands. The authors identify temperature, salinity, and chlorophyll a as key drivers influencing $N_2O$ variability, using machine learning approaches to analyze their relationships. Their findings contribute valuable data on $N_2O$ fluxes in a Mediterranean coastal system, an area where such measurements remain limited. However, several aspects of the study warrant further clarification and discussion. Below are my comments and suggestions for this manuscript.

The title suggests that the study identifies drivers of the spatiotemporal distribution of dissolved $N_2O$ and air-sea exchange. However, the manuscript does not clearly demonstrate how the measured parameters (drivers) directly influence these variations over time and space. Additionally, why were these specific parameters chosen as drivers? For instance, why was $NH_4^+$ not included.

In addition, the study area appears to be relatively shallow. Were sediment $N_2O$ fluxes considered? If not, could the lack of sediment contribution explain the weak relationships observed in the study?

Line 21: Consider providing the average air-sea flux with standard deviation for clarity and comparison.

Line 36: Briefly introduce nitrification and denitrification, including the conditions under which these processes occur.

Line 86: Time sampling timeline is unclear. Please specify when the samples were collected.

Line 136 - 143: Please specify the bottle type, volume, and collection frequency for DO, Chl a, nutrient, DOC and $N_2O$ samples.

Line 202: This section lacks descriptions of basic environmental parameters, such as surrounding nutrient and oxygen variations. While these results are presented in Section 3.2, that section mainly focuses on their impact on $N_2O$ rather than describing the environmental parameters themselves. To improve clarity and provide a better understanding of local environmental conditions, consider including these descriptions here

Line 205: Is it BP or PB? The abbreviation appears inconsistent. Please check for consistency, including in the Methods section.

Line 226-234: The table is unclear. It states significant seasonal, yearly, monthly, or station differences for some parameters, but these are not evident in Table 1. Please clarify or update the table accordingly.

Line 254: Unit?

Line 265: The not shown results could be included in the supplementary material.

Line 285: The study found low $NO_3^-$ and $NO_2^-$ concentrations, weak nitrification signals, and no correlation between AOU and $N_2O$. Based on these findings, the authors propose that photosynthetic organisms-driven NO reduction could be a source of $N_2O$ in their system. This presents an interesting alternative pathway to nitrification and denitrification. However, based on these data, this hypothesis remains uncertain. Please provide a more detailed explanation of how this mechanism works.

Line 296: The explanation in Section 3.2 is largely based on the GBM and CVB results; however, it lacks a deeper discussion on the relationships between parameters and $N_2O$, as well as a literature review for comparison. Expanding on these aspects would strengthen the section.

Line 312: I agree that coastal air-sea $N_2O$ exchange is an important parameter in the context of GHG emissions. However, this section lacks comparisons with similar studies or coastal systems, making the discussion somewhat limited.

Lines 329-339: The description of the study sites is insufficient. It is unclear which areas are open regions and which have seagrass. This lack of detail in the study location section makes their sudden appearance in the discussion feel abrupt. In addition, please explain the mechanism and process by which vegetation acts as a sink for $N_2O$.

Line 340: Please provide references for European seagrass constitute.

Line 344: The Methods section does not describe the presence, extent, or coverage of seagrass meadows at the study sites. Since seagrass is later discussed as a factor influencing $N_2O$ dynamics, please provide details on seagrass distribution and area to support these claims.

Line 353: The estimation of $N_2O$ GWP is based on a 1 km offshore distance and 1,428 km of coastal length. What is the basis for this area calculation? Additionally, uncertainty should be provided for this conversion. Does this area calculation include both seagrass meadows and bare sediments?

Line 361: Line 361: Ensure unit consistency—sometimes 'nM' is used, while other times 'nmol $L^{-1}$' appears. Please standardize throughout the text

There seems to be inconsistent use of r and $R^2$. Please ensure consistent notation throughout the text.

---

## Author Comment (AC1)

This manuscript set out to evaluate spatial and temporal variability of surface $N_2O$ concentrations and estimates of air-sea flux as well as evaluate the drivers of $N_2O$ concentration and air-sea flux from three sites located among the Balearic Islands Archipelago. The monitoring period was relatively extensive ranging from 2018 to 2023, and as far as I know these are among the first observations of $N_2O$ concentrations and estimations of air-sea flux from these specific sites.

The authors found that $N_2O$ concentrations were mainly driven by seasonal temperature fluctuations, with no significant site or yearly differences. The system description with respect to $N_2O$ concentration and estimated fluxes, along with length of the dataset warrant publishing this manuscript, but the manuscript needs some additional analyses and polishing as there are some issues that need to be addressed before publication.

We appreciate the reviewer for her/ his constructive and thoughtful comments, which have enabled us to enhance the clarity and scientific quality of the manuscript. We have thoroughly addressed each of the points raised and made the necessary changes in the revised version of the manuscript. Below, we offer a detailed, point-by-point response to each comment. All changes in the manuscript are highlighted using track changes for easy reference.

**General Comments:**
1. Please include the sampling depth in the description of the study area. From the onset, it was hard to contextualize results from the different sampling sites without understanding the depth water was taken from in the Bay of Palma. It is unclear if Bay of Palma samples were also collected from the buoy sensor depth as the only mention of $N_2O$ sample collection depth is at the bottom of the second paragraph in the section (line 106) which discusses sampling depth at the bay of Santa Maria site. Moreover, this description is separated from the description of the Bay of Palma sampling (1st paragraph in the section), so the authors might also consider indicating sampling / sensor depth in the first paragraph and on Figures 2 and 3.

We thank the reviewer for this valuable suggestion. We have clarified the sampling depth for each site in the "Study Area" section (lines 99, 106 and 117). Specifically, we now indicate that both the sensor and water samples at the Palma Bay (PB) were collected at a depth of approximately 1 m. Similarly, we have reiterated the sampling depth of 4 m at the Bay of Santa Maria (CA) and specified that the sampling depth at Cape Ses Salines (CS) is approximately 0.5 m. We also clarified that water samples were collected at the same depth as the sensors at all three stations. Additionally, we will update Figures 2 and 3 footnotes to indicate the sampling and sensor depths as suggested.

2. Quantifying air-sea flux in many marine $N_2O$ studies is often undertaken using parameterizations of gas transfer velocity ($k$), in the absence of direct measurements on site. Commonly these parameterizations include wind speed, but in some instance's other variables such as current speed, heat flux, or even rainfall can be used. The $k$ parameterization used by the authors in this manuscript was described in Dobashi and Ho (2023) and was developed using $^3He/SF_6$ tracer release in a shallow (<3.5m) seagrass bed in Florida Bay, USA. This certainly seems to be an appropriate parameterization for two of the three study sites in this manuscript as they were located in relatively shallow water (Bay of Santa Maria site ~8m deep; Cape Ses Salines lighthouse site 2m deep). The other site (Bay of Palma) however, was described as being approximately 30m deep, so Dobashi and Ho (2023) may not be the most appropriate parameterization here. I would recommend including at least 2 additional parameterizations used for deeper (> 10m) coastal areas in Table 2 for this site.

We appreciate the reviewer's insightful observation. Following the suggestion, we have now included two additional gas transfer velocity parameterizations that are commonly used in deeper coastal waters: Wanninkhof (2014) and Cole & Caraco (1998). These parameterizations were applied to all three sites to enable comparison, but they are particularly relevant for the Bay of

Palma (PB) station, where the water depth reaches approximately 30 m. The description of these additional parameterizations has been incorporated into the "Flux estimation and other calculations" Lines 192-195 of the Methods section. Additionally, the resulting annual fluxes calculated using all three parameterizations are now included in Table 2.

3. I think the examination of the drivers of [$N_2O$] would gain context by examining drivers of $N_2O$% saturation as well. It is well known that dissolved gas concentrations are driven by temperature, but I believe a more useful question is what is driving $N_2O$% saturation in coastal systems. Examining the drivers of $N_2O$ % Saturation should be included before publication. I do not see the results any analysis investigating drivers of $N_2O$ in this manuscript and the explanation of the analysis is made vague by the use of the term "$N_2O$ levels" (line 181, line 288, Figure 4) and "$N_2O$" (line 190) instead of more precise terms "$N_2O$ concentrations" or "$N_2O$ % Saturation". I would like to see this, and other instances of vague language (*e.g.* "$N_2O$ variability" line 283), clarified before publication. Also, I would like to see $N_2O$ % Saturation included in Figure 3.

We appreciate the reviewer for this constructive suggestion. We completely agree that analyzing the factors influencing $N_2O$ saturation percentage offers valuable ecological context, especially in dynamic coastal systems where various physical and biogeochemical processes impact gas exchange and concentrations.

In response, we conducted an exploratory analysis to assess the potential drivers of $N_2O$ % saturation (Sat%$N_2O$). The results were indeed consistent with those observed for $N_2O$ concentrations. Specifically, temperature and salinity emerged as the dominant explanatory variables, with relative importance values of 41.9% and 15.7%, respectively. The remaining environmental predictors ($NO_3^-$, $NO_2^-$, DOC, Chl-a) exhibited comparable secondary contributions (see the table below).

| Feature | Importance |
|---------|------------|
| Temperature | 41.865173 |
| Salinity | 15.659371 |
| NO3 | 12.119104 |
| Chla | 11.971512 |
| DOC | 10.928545 |

While $N_2O$ % saturation provides insights into the degree of equilibrium with the atmosphere and is highly relevant for interpreting air–sea fluxes, we chose to focus the primary analysis on absolute $N_2O$ concentrations for comparability with literature: Most published studies examining spatial and seasonal trends, as well as biogeochemical drivers of $N_2O$ in marine systems, report absolute concentrations, facilitating meaningful comparisons.

That said, we agree that the inclusion of saturation metrics strengthens the ecological interpretation of our findings. Therefore, to address the reviewer's suggestions, we have:
• Clarified terminology throughout the manuscript by replacing vague or ambiguous terms such as "$N_2O$ levels" or "$N_2O$ variability" with precise descriptors, e.g., "$N_2O$ concentration" and "$N_2O$ % saturation" where appropriate.
• Included $N_2O$ % saturation in Figure 3, to visualize its spatiotemporal patterns alongside $N_2O$ concentrations, thereby offering a more holistic representation of the system's $N_2O$ dynamics.

We believe these adjustments enhance both the clarity and scientific rigor of the manuscript while preserving a focused analytical framework.

4. Quantification of uncertainty should also be addressed, especially considering $N_2O$ observations are made from duplicate samples. Please specify how uncertainty was propagated

through the estimations of air-sea flux (uncertainty can come from many sources including variability between replicates, wind speed variability, temperature variability etc.).

We thank the reviewer for pointing out the importance of uncertainty quantification. In the revised manuscript, we have included an estimation of the propagated uncertainty in the calculation of air-sea $N_2O$ fluxes, based on the variability between duplicate $N_2O$ measurements. This approach accounts for the analytical uncertainty associated with the gas concentration measurements, which is the main experimental source of variability in our dataset.

In addition, we recalculated annual $N_2O$ fluxes using a trapezoidal integration approach over time, which—although it did not significantly alter the magnitude of the annual flux estimates— provides a more robust and continuous representation of seasonal variability. Both the uncertainty propagation method and the revised flux calculation approach are now described in the Methods section.

In this vein, there should be some indication about which $N_2O$ % saturation observations are and are not significantly different than 100% (should be indicated in Table 1). What appears to be slight under saturation at sites CA and CS may not be significantly lower than 100%, but if they are I think this warrants some additional discussion about what might be driving this. The same should be indicated for air-sea fluxes shown in Figure 6 (which of these are significantly greater than 0, and which are just artefacts of statistical noise?)

Thank you for the suggestion. We have now conducted one-sample t-tests to evaluate whether the observed $N_2O$ % saturations are significantly different from 100%. This information is reported in the footnote of Table 1. We also applied the same statistical approach to air–sea $N_2O$ fluxes. Fluxes that are significantly different from zero are indicated in the legend of Figure 6. These clarifications allow for a more robust interpretation of undersaturation or net fluxes.

**Specific Comments:**
Line 40: Not all nutrient inputs are "solid", recommend removing this word and retain as "land-derived nutrient inputs".

Thank you for the suggestion. We agree that not all land-derived nutrient inputs are solid. We have revised Line 40 accordingly and now refer to them simply as "land-derived nutrient inputs".

Line 43: recommend rephrasing "which also stimulates the generation of $N_2O$" to "which can stimulate the generation of $N_2O$".

Thank you for your suggestion. We have updated the sentence to: "which can stimulate the generation of $N_2O$" for a more cautious interpretation.

Line 91: What is the sensor depth?

The sensor depth is now specified in the Methods section.

Line 155: Need to show airport wind station on the map on Figure 1.

Thank you for pointing this out. We have added the location of the airport wind station to Figure 1.

Line 204: Have you defined which months are indicated by "summer", "spring", "winter", "autumn"? Is this "June to Aug" or "July to Sep"? Please define.

We now clarify the seasonal definitions in the text: *"winter" refers to December–February, "spring" to March–May, "summer" to June–August, and "autumn" to September–November.*

Line 362: "N$_2$O variability" is vague. Need to explicitly say what metric of N$_2$O is being described: "variability of N$_2$O concentration" or "variability of N$_2$O % saturation".

Thank you for the comment. The sentence has been revised to specify the metric being described: "variability of N$_2$O concentration".

Line 366: I think the term "impacted" should also be cautiously used here. There are some major depth differences between the stations, and I assume seagrass coverage differences as well.

We agree that "impacted" may be too strong or ambiguous in this context. We have replaced it with "influenced", which we believe better reflects the range of possible environmental factors, including depth and seagrass coverage differences.

Line 366: Without showing which estimates of N$_2$O flux are significantly greater than 0, it is hard to justify saying the sites are weak sources of N$_2$O to the atmosphere. They could well be in equilibrium.

We acknowledge this concern and have revised the text to avoid overinterpreting the flux estimates. We now note that *"some sites appear to be weak sources of N$_2$O, although many fluxes are close to equilibrium and not significantly different from zero."*

**Technical Corrections:**
Line 31: Please include a reference for this sentence.

 A reference has been added to support this sentence.

Line 51: Please include a reference for this sentence.

A reference has been included for this sentence as well.

Line 74: "N$_2$O" has already been defined as "nitrous oxide". Just use "N$_2$O".

 Line 103: "N$_2$O" has already been defined as "nitrous oxide". Just use "N$_2$O".

Line 117: "N$_2$O" has already been defined as "nitrous oxide". Just use "N$_2$O".

**Lines 74, 103, 117:**

As suggested, we have replaced the repeated full name "nitrous oxide" with "N$_2$O" in all instances after the term was first defined.

---

## Author Comment (AC2)

Reviewer's comments for Drivers of the spatiotemporal distribution of dissolved nitrous oxide and air-sea exchange in a coastal Mediterranean area

This study investigates the spatiotemporal distribution of dissolved $N_2O$ and air-sea exchange in the coastal waters of the Balearic Islands. The authors identify temperature, salinity, and chlorophyll a as key drivers influencing $N_2O$ variability, using machine learning approaches to analyze their relationships. Their findings contribute valuable data on $N_2O$ fluxes in a Mediterranean coastal system, an area where such measurements remain limited. However, several aspects of the study warrant further clarification and discussion. Below are my comments and suggestions for this manuscript.

We sincerely thank the reviewer for the thorough and constructive evaluation of our manuscript "*Drivers of the spatiotemporal distribution of dissolved nitrous oxide and air-sea exchange in a coastal Mediterranean area.*" We appreciate the recognition of the novelty and relevance of our contribution, particularly regarding the scarce $N_2O$ observations in the Mediterranean Sea. We carefully considered all comments and suggestions, which have led to significant improvements in the manuscript.

In the revised version, we have clarified the methodological approach, strengthened the discussion on mechanistic drivers of $N_2O$ variability, included supporting literature, and provided more precise information about sampling design, environmental context, and site-specific features such as the presence of seagrass meadows. Below, we provide point-by-point responses to each of the reviewer's comments, including detailed explanations of the revisions made and the corresponding locations within the manuscript.

The title suggests that the study identifies drivers of the spatiotemporal distribution of dissolved $N_2O$ and air-sea exchange. However, the manuscript does not clearly demonstrate how the measured parameters (drivers) directly influence these variations over time and space. Additionally, why were these specific parameters chosen as drivers? For instance, why was $NH_4^+$ _not included.

We thank the reviewer for highlighting these critical points. We acknowledge the importance of clearly justifying both the selection of environmental parameters and their relationship to the observed $N_2O$ dynamics across time and space.

To identify the most influential variables, we first conducted a broad exploratory analysis that included a wide range of physical, chemical, and biological parameters measured during the study—such as $NH_4^+$, $PO_4^{3-}$, turbidity, and others. We applied variable importance metrics (e.g., based on random forest models and correlation analyses) and tested multiple model configurations to evaluate their predictive strength and consistency across sites and seasons.

This data-driven approach resulted in the selection of a final set of variables that consistently explained the greatest variation in dissolved $N_2O$ concentrations and saturation. These include:
• Temperature and salinity, which influence gas solubility and microbial kinetics;
• $NO_3^-$ and $NO_2^-$, as key substrates or intermediates in nitrification and denitrification;
 • DOC, which supports heterotrophic microbial activity;
• Chlorophyll *a serves* as a proxy for primary production and potential oxygen variability, as well as for organic matter.

Although $NH_4^+$ was initially considered, its influence in the model was relatively low, and in some cases, its inclusion reduced the overall predictive power due to missing data or noise in certain sites. Therefore, it was excluded from the final analysis to ensure robustness and comparability across the entire spatiotemporal dataset. We have now clarified this process in the revised

Methods section Lines 223-225 and expressly acknowledge the exclusion of $NH_4^+$ as a limitation and an opportunity for future, targeted studies.

Additionally, we have revised the Results and Discussion to explain more explicitly the mechanistic pathways through which the retained variables influence $N_2O$ production, accumulation, and fluxes. This includes:
• The role of temperature and salinity in modulating solubility and stratification; Lines 312-325.
• How $NO_3^-$ enrichment reflects anthropogenic inputs and potential for enhanced nitrification; Lines 345-348.
• And the contribution of DOC and Chl-a to oxygen availability and microbial dynamics. Lines 336-342.

Finally, we have ensured that the title and narrative more clearly align with the structure of the analysis, emphasizing the identified environmental associations with $N_2O$ distributions.

In addition, the study area appears to be relatively shallow. Were sediment $N_2O$ fluxes considered? If not, could the lack of sediment contribution explain the weak relationships observed in the study?

We appreciate the reviewer's pertinent observation regarding potential benthic contributions to the $N_2O$ budget, particularly in shallow coastal areas such as those studied here.

At the time of this study, no direct measurements of sediment–water $N_2O$ fluxes were available, as the required instrumentation (e.g., benthic chambers or microprofiling sensors) and logistical support were not yet in place. For this reason, our analysis focused exclusively on water column $N_2O$ dynamics and air–sea exchange.

We fully acknowledge that sediment-derived $N_2O$ fluxes may play a role in shaping dissolved $N_2O$ concentrations, especially under low-oxygen or organic-rich conditions, and that the absence of these data may contribute to the moderate explanatory power of some of our observed relationships. However, the predominantly oxygenated conditions observed throughout the study period, along with negative AOU values, could suggest limited sedimentary N2O production in our sites.

Importantly, we are currently conducting a dedicated follow-up study designed to quantify benthic $N_2O$ fluxes using sediment core incubations and in situ benthic chambers in the same sampling locations. These measurements will allow us to better assess the relative contribution of sediment processes to the coastal $N_2O$ budget in future analyses.

We have now included a sentence in the Results and Discussion section to acknowledge this limitation and highlight the importance of benthic flux measurements in future efforts. Lines 346-348.

Line 21: Consider providing the average air-sea flux with standard deviation for clarity and comparison.

We thank the reviewer for this helpful suggestion. As recommended, we have included the average air–sea $N_2O$ flux along with its standard deviation in the Abstract to enhance clarity and facilitate better comparison with other studies. Line: 22

Line 36: Briefly introduce nitrification and denitrification, including the conditions under which these processes occur.

We appreciate the reviewer's insightful suggestion. In response, we have added a concise overview of the nitrification and denitrification processes, including the environmental conditions

in which they take place, to the Introduction section (Lines: 38-42). This enhancement improves the contextual understanding of the microbial pathways that lead to $N_2O$ production in marine systems.

Line 86: Time sampling timeline is unclear. Please specify when the samples were collected.

We thank the reviewer for pointing this out. The sampling timeline has now been clarified in Section 2.1 – Study Area, where we specify the exact time periods and seasonal coverage of the data collection.

Line 136 - 143: Please specify the bottle type, volume, and collection frequency for DO, Chl $a$, nutrient, DOC and N2O samples.

We appreciate the reviewer's attention to methodological detail. As suggested, we have now added the bottle type, volume, and collection frequency for DO, Chl $a$, nutrients, and DOC in the Materials and Methods section. For $N_2O$, we considered that all relevant sampling information, including handling and analytical procedures, was already sufficiently detailed in the original submission.

Line 202: This section lacks descriptions of basic environmental parameters, such as surrounding nutrient and oxygen variations. While these results are presented in Section 3.2, that section mainly focuses on their impact on $N_2O$ rather than describing the environmental parameters themselves. To improve clarity and provide a better understanding of local environmental conditions, consider including these descriptions here.

We thank the reviewer for this valuable suggestion. As recommended, we have now added a brief summary of key environmental parameters—including oxygen, nitrate, nitrite, and DOC concentrations—at the end of Section 3.1 (Lines 261-269). This addition provides a clearer and more complete overview of the local environmental conditions, independently of their influence on $N_2O$ dynamics, which are further explored in Section 3.2.

Line 205: Is it BP or PB? The abbreviation appears inconsistent. Please check for consistency, including in the Methods section.

We apologize for this inconsistency. The station abbreviation has been thoroughly reviewed and corrected throughout the manuscript, including the Methods section, to ensure consistent usage.

Line 226-234: The table is unclear. It states significant seasonal, yearly, monthly, or station differences for some parameters, but these are not evident in Table 1. Please clarify or update the table accordingly.

We appreciate the reviewer's observation. To avoid confusion, we have removed references to Table 1 where statistical differences were not clearly supported. Additionally, we have revised and clarified Table 1 to indicate significant differences where applicable explicitly. For example, we have now marked when $N_2O$ % saturation values significantly differ from 100%, as suggested in a previous comment, to improve clarity and interpretability.

Line 254: Unit?

We apologize for the oversight. The unit has now been included in the revised manuscript to ensure clarity.

Line 265: The not shown results could be included in the supplementary material.

We thank the reviewer for this observation. Due to the strong visual and statistical similarity between this relationship and the one already shown in Figure 5 (temperature vs. Sat%$N_2O$), we considered that including an additional figure or supplement would be redundant. For this reason, we have chosen to report the statistical results directly in the text, providing the exact correlation strength and significance for transparency.

Line 285: The study found low $NO_3^-$ _and $NO_2^-$ _concentrations, weak nitrification signals, and no correlation between AOU and $N_2O$. Based on these findings, the authors propose that photosynthetic organisms-driven NO reduction could be a source of $N_2O$ in their system. This presents an interesting alternative pathway to nitrification and denitrification. However, based on these data, this hypothesis remains uncertain. Please provide a more detailed explanation of how this mechanism works.

We appreciate the reviewer for this insightful comment. We concur that the hypothesis related to $N_2O$ production through NO reduction by photosynthetic organisms is indeed speculative in our study, given the indirect evidence presented. Our goal was not to claim this as the primary pathway, but to point out that it could be a viable complementary mechanism to consider under the specific conditions observed—particularly, low $NO_3^-/NO_2^-$ availability, well-oxygenated waters, and the seasonal correlation between $N_2O$ and Chl $a$ peaks.

To clarify this mechanism, we have expanded the explanation in the Results and Discussion section (Lines: 348-362) . Briefly, recent studies have shown that certain photosynthetic microorganisms, including marine phytoplankton and cyanobacteria, can reduce intracellular nitric oxide (NO) to $N_2O$ as part of a detoxification or signaling pathway (e.g., Burlacot et al., 2020). This pathway may be more active under high-light conditions or when nitrite accumulates transiently, leading to intracellular NO production. Although our data do not directly capture these intracellular processes, the temporal alignment between Chl $a$ and $N_2O$ concentrations, combined with the lack of evidence for pelagic nitrification, supports the consideration of this emerging mechanism in productive surface waters.

We have revised the manuscript to clarify the tentative nature of this interpretation and to improve the description of the underlying mechanism and its supporting literature.

Line 296: The explanation in Section 3.2 is largely based on the GBM and CVB results; however, it lacks a deeper discussion on the relationships between parameters and $N_2O$, as well as a literature review for comparison. Expanding on these aspects would strengthen the section.

We thank the reviewer for this helpful suggestion. As recommended, we have expanded Section 3.2 to include a more detailed discussion on the mechanistic relationships between $N_2O$ concentrations and the main environmental variables identified (e.g., temperature, salinity, Chl $a$, DOC, $NO_3^-$), as well as their potential interactive effects. We have also incorporated relevant literature for comparison with other coastal and estuarine systems. These additions can be found in the revised manuscript in lines 384-392.

Line 312: I agree that coastal air-sea $N_2O$ exchange is an important parameter in the context of GHG emissions. However, this section lacks comparisons with similar studies or coastal systems, making the discussion somewhat limited.

We value the reviewer's perceptive feedback. In response, we have updated Section 3.3 to include comparisons with other coastal and estuarine systems from around the globe. This update specifically includes reference values from recent studies on $N_2O$ fluxes in various environments, including both vegetated and non-vegetated shallow coastal areas, estuaries, and seagrass meadows. These enhancements provide context for the magnitude and direction of the observed fluxes in the Balearic Sea and reinforce the discussion about the regional and global significance of our findings. Please see lines 421-426 and 443-446 in the updated manuscript.

Lines 329-339: The description of the study sites is insufficient. It is unclear which areas are open regions and which have seagrass. This lack of detail in the study location section makes their sudden appearance in the discussion feel abrupt. In addition, please explain the mechanism and process by which vegetation acts as a sink for $N_2O$.

Thank you for pointing this out. We have revised Section 2.1 to include more precise descriptions of the physical and benthic characteristics of each sampling site. Specifically, we now clarify that PB is located in an open embayment without seagrass influence, CA is surrounded by dense Posidonia oceanica meadows, and CS includes shallow waters with patchy vegetation. This information has been added at the end of Section 2.1 (Lines 122-128), ensuring a smoother transition and improved contextual understanding of the discussion that follows in later sections.

We agree with the reviewer and have now included a paragraph in Section 3.3 (Lines 437-442) explaining the mechanisms by which seagrass meadows can act as sinks for $N_2O$. This includes enhanced denitrification in oxygenated sediments, reduced nitrogen availability due to high primary production, and microbial community processes favored by seagrass root systems. This addition supports our interpretation of the lower fluxes observed at vegetated sites and provides stronger mechanistic insight.

Line 340: Please provide references for European seagrass constitute.

Thank you for the suggestion. We have now added a reference to substantiate the statement regarding the contribution of European seagrasses to the global extent. The reference has been included in Section 3.3 (Line 448), supporting the ecological relevance of Mediterranean seagrass meadows in the context of coastal $N_2O$ fluxes.

Line 344: The Methods section does not describe the presence, extent, or coverage of seagrass meadows at the study sites. Since seagrass is later discussed as a factor influencing $N_2O$ dynamics, please provide details on seagrass distribution and area to support these claims.

Thank you for your valuable observation. In response, we have updated Section 2.1 (Study area) to explicitly describe the presence and ecological context of seagrass meadows at each station. Specifically, we now state that the CA site is located within a dense Posidonia oceanica seagrass meadow, the CS site includes shallow waters with patchy vegetated coverage, and the PB station is situated in an open bay without seagrass. These additions clarify the relevance of seagrass presence in the interpretation of $N_2O$ dynamics and improve the connection between site characteristics and the discussion provided in Section 3.3.

Line 353: The estimation of $N_2O$ GWP is based on a 1 km offshore distance and 1,428 km of coastal length. What is the basis for this area calculation? Additionally, uncertainty should be provided for this conversion. Does this area calculation include both seagrass meadows and bare sediments?

We appreciate the reviewer's thoughtful observation. The 1 km offshore distance combined with the 1,428 km of coastal length was used as a first-order approximation of the potentially active $N_2O$ exchange area in the Balearic coastal zone. This estimation is based on previous methodological approaches applied in similar large-scale GHG assessments, including Resplandy et al. (2024) and Rosentreter et al. (2023), which use coastal buffer zones (typically 1–3 km) to estimate regional coastal fluxes.

While this approach does not differentiate between vegetated (seagrass) and non-vegetated (bare sediment) areas, it provides a conservative estimate of the potential footprint for $N_2O$ exchange in the region. We have clarified this point in Section 3.3, noting that this surface likely

encompasses a mix of substrates, and that more refined spatial mapping would be required for precise habitat-specific flux quantification.

Additionally, the uncertainty of the GWP estimate is reported ($\pm 5.7 \times 10^{-7}$ Pg $CO_2$-eq $y^{-1}$) and reflects variability in flux measurements across sites and time.

Line 361: Line 361: Ensure unit consistency—sometimes 'nM' is used, while other times 'nmol $L^{-1}$' appears. Please standardize throughout the text

Thank you for pointing this out. We have reviewed the manuscript thoroughly and standardized the units, using 'nmol $L^{-1}$' consistently throughout the text to ensure clarity and uniformity.

There seems to be inconsistent use of r and R2. Please ensure consistent notation throughout the text.

Thank you for your careful observation. We have reviewed the manuscript and standardized the correlation notation throughout the text.

---

## Author Response (AR2)

Dear Reviewers,

We sincerely appreciate your constructive comments, which have aided us in enhancing the quality and clarity of the manuscript. Below, please find our responses to your specific suggestions:

**Reviewer 1:**

*Comment: The figure 6 caption states that this is a box plot; however, the figure itself seems to be a bar chart instead of a box plot. Please revise the caption to accurately describe the type of figure.*

**Response:** Thank you for pointing this out. We have revised the caption of Figure 6 to accurately describe the figure as a bar chart, rather than a box plot.

**Reviewer 2:**

*Comment: I would like to see explicit annotation of which specific fluxes are not significantly different from equilibrium in the figure.*

**Response:** Thank you for your suggestion. When analyzing individual $FN_2O$ flux measurements, all observations were found to be not significantly different from zero ($p > 0.05$) at all stations and sampling dates. This indicates that, although the mean fluxes per station were statistically different from zero, individual flux measurements exhibited high variability and were indistinguishable from null fluxes during the sampling periods. This information has been included in the caption of Figure 6, the Abstract, the Results section, and the Conclusions to clarify this point.

We trust that these revisions satisfactorily address your concerns. Thank you once more for your valuable feedback.

Sincerely,

Susana Flecha

On behalf of all co-authors